A taxonomic revision of Rhizophora L. (Rhizophoraceae) in Thailand

Ngernsaengsaruay Chatchai 1 2 fsciccn@ku.ac.th
http://orcid.org/0009-0001-7325-6109 Chanton Pichet 3
Leksungnoen Nisa 4
Uthairatsamee Suwimon 4
Mianmit Nittaya 5
1 Department of Botany, Faculty of Science, Kasetsart University , Chatuchak, Bangkok , Thailand
2 Biodiversity Center Kasetsart University (BDCKU) , Chatuchak, Bangkok , Thailand
3 Suan Luang Rama IX Foundation , Nong Bon Subdistrict, Prawet District, Bangkok , Thailand
4 Department of Forest Biology, Faculty of Forestry, Kasetsart University , Chatuchak, Bangkok , Thailand
5 Department of Forest Management, Faculty of Forestry, Kasetsart University , Chatuchak, Bangkok , Thailand
Casazza Gabriele
Electronic publication date: 2024 Jun 28
Publication date: 2024
Volume: 12
Electronic Location ID: e17460
Received 2024 Jan 25; Accepted 2024 May 3
Copyright: © 2024 Ngernsaengsaruay et al.
Copyright year: 2024
Copyright holder: Ngernsaengsaruay et al.
License: This is an open access article distributed under the terms of the Creative Commons Attribution License, which permits unrestricted use, distribution, reproduction and adaptation in any medium and for any purpose provided that it is properly attributed. For attribution, the original author(s), title, publication source (PeerJ) and either DOI or URL of the article must be cited.
License URL: https://creativecommons.org/licenses/by/4.0/

Keywords: Leaf anatomy, Lectotypification, Malpighiales, Morphology, Neotype, Palynology, Rhizophoreae, True mangrove trees, Vivipary

Funding: Biodiversity Center Kasetsart University (BCKU) International SciKU Branding (ISB), Faculty of Science, Kasetsart University This study was funded by Kasetsart University from research funding through the Biodiversity Center Kasetsart University (BCKU) and the International SciKU Branding (ISB), Faculty of Science, Kasetsart University. The funders had no role in study design, data collection and analysis, decision to publish, or preparation of the manuscript.

==============================
A taxonomic revision of Rhizophora L. (Rhizophoraceae) in Thailand is presented. Two species, R. apiculata Blume and R. mucronata Poir., are enumerated with updated morphological descriptions, illustrations and a taxonomic identification key, together with notes on distributions, habitats and ecology, phenology, conservation assessments, etymology, vernacular names, uses, and specimens examined. Three names in Rhizophora, are lectotypified: R. apiculata and two associated synonyms of R. mucronata (i.e., R. latifolia Miq. and R. macrorrhiza Griff.). R. longissima Blanco, a synonym of R. mucronata, is neotypified. All two Rhizophora species have a conservation assessment of Least Concern (LC). Based on the morphological identification, these two species can be distinguished from one another by the shape and width of the leaf laminae and the length of a terminal stiff point of the leaf laminae; the type and position of the inflorescences and the number of flowers per inflorescence; the character and color of the bracteoles; the presence or absence of the flower pedicels; the shape of the mature flower buds; the shape, color, and texture of the sepals; the shape, character, and the presence or absence of hairs of the petals; the number of stamens per flower; the size of the fruits; the color and size of the hypocotyls; the color and diameter of the cotyledonous cylindrical tubes; and the color of the colleters and exudate. The thick cuticles, sunken stomata, large hypodermal cells, and cork warts are adaptive anatomical features of leaves in Rhizophora that live in the mangrove environments. The pollen grains of Thai Rhizophora species are tricolporate, prolate spheroidal or oblate spheroidal shapes, small-sized, and reticulate exine sculpturing.

Introduction

Rhizophoraceae Pers. are well known as the richest mangrove family, having four exclusively mangrove genera and 18 species: Bruguiera Lam. ex Savigny (five species), Ceriops Arn. (five species), Kandelia (DC.) Wight & Arn. (two species), and Rhizophora L. (six species) (Tomlinson, 1986, 2016; Sheue, Chen & Yang, 2012; POWO, 2023). The family is diversified both ecologically and morphologically and has 15 genera with 137 species (Sheue, Chen & Yang, 2012; POWO, 2023). All members of the Rhizophoraceae have large and conspicuous interpetiolar, glabrous and caducous stipules on both vegetative and reproductive shoots, strongly ensheathing the young leaves and inflorescences (Hou, 1958). Several to hundreds of aggregated finger-like colleters mixed with milky mucilage can be observed with the naked eye at the adaxial base of stipules in the species of mangrove Rhizophoraceae (Tomlinson, 1986, 2016; Sheue, 2003). Except for the genus Bruguiera, colleters, although few in number, are also present within the bracteoles of mangrove Rhizophoraceae inflorescences (Sheue, 2003; Sheue, Chen & Yang, 2012). The structural and mechanical protection provided by stipules and colleter exudates may help shield the young shoots of these mangrove plants from their harsh environments (Sheue, Chen & Yang, 2012).

The tribe Rhizophoreae Bartl. is characterised by the following: true mangrove shrubs or trees, with aerial roots; glabrous and caducous interpetiolar stipules, sheathing the terminal bud and the young leaves; decussate and entire leaves; cymose inflorescences; 4–16-merous flowers; mostly diplostemonous (with twice as many stamens as petals); half to fully inferior ovaries, 2–3-carpellate with two ovules per carpel; baccate and fibrous fruits, usually 1-seeded; viviparous germination. This pantropical tribe has four genera (Bruguiera, Ceriops, Kandelia, and Rhizophora (Henslow, 1878; King, 1897; Hou, 1958; Schwarzbach, 2014)).

Generally, mangrove plant species categorized as species that are exclusively limited to the mangrove environments (referred as true mangrove, strict mangrove or obligate mangrove) and nonexclusive species that are mainly distributed in a terrestrial or aquatic habitats but also occur in the mangrove ecosystem (referred as mangrove associate, semi-mangrove or back mangrove) (Tomlinson, 1986, 2016; Parani et al., 1998; Lacerda et al., 2002; Wang et al., 2011). True mangroves possess all or most of the following features: (1) occurring only in mangrove environments and not extending into terrestrial communities, (2) a major role in the structure of the mangrove community, sometimes forming pure stands, (3) morphological specializations for the mangrove environment (e.g., aerial roots, vivipary), (4) some physiological mechanism for salt exclusion and/or salt excretion, and (5) taxonomic distinctiveness from terrestrial relatives. Species found in mangrove environments that do not possess all these characteristics are categorized as mangrove associates (Tomlinson, 1986, 2016). The term “mangrove associates” was coined for flora representing non-arborescent, herbaceous, sub-woody and climber species that are found to grow mostly in regions adjoining the tidal periphery of mangrove habitats (Mepham & Mepham, 1985; Chanda et al., 2016). True mangroves are considered to be facultative or obligate halophytes (true halophytes) whereas mangrove associates are those which can be classified as glycophytes with certain salt tolerance (Wang et al., 2011; Chanda et al., 2016; Quadros & Zimmer, 2017).

Rhizophora is a group of tropical and subtropical mangrove trees, or collectively called true mangroves and is a small genus belongs to the order Malpighiales Juss. ex Bercht. & J. Presl, the family Rhizophoraceae (The Angiosperm Phylogeny Group, 2016) and the tribe Rhizophoreae (Schwarzbach, 2014). The generic name Rhizophora is derived from the Greek compound words, “rhiz-”, “rhizo-” meaning root, and “-phoros”, “-phorus” meaning -bearing or carrying, referring to the stilt roots or prop roots (Stearn, 1992; Radcliffe-Smith, 1998; Gledhill, 2002). Eleven taxa have been recognized including six accepted species and five hybrids, of which two species and one hybrid in America, three species and one hybrid in Africa, three species and two hybrids in the Indian subcontinent, China, and Southeast Asia (POWO, 2023), three species and one hybrid in Australia (Duke & Bunt, 1979; McCusker, 1984; POWO, 2023), and four species and three hybrids in Pacific (Micronesia, Melanesia, and Polynesia) (Tomlinson, 1978; POWO, 2023). These three Indian subcontinent, China, and Southeast Asian species are Rhizophora apiculata Blume, R. mucronata Poir., and R. stylosa Griff. (Hou, 1958, 1960; Qin & Boufford, 2007; Marchand, 2008; POWO, 2023), plus two hybrids: R. × annamalayana Kathiresan (R. apiculata × R. mucronata) and R. × lamarckii Montrouz. (R. apiculata × R. stylosa) (Parani et al., 1997; Ng & Chan, 2012; Ng, Chan & Szmid, 2013; Setyawan, Ulumuddin & Pandisamy, 2014; Ragavan et al., 2015; POWO, 2023).

Previously Hou (1970) published the genus Rhizophora in the Flora of Thailand and two species were recognized, R. apiculata and R. mucronata, but this report was short in morphological descriptions and other information. In this article, we had extensively examined Thai specimens of Rhizophora in various local and international herbaria including virtual herbarium repositories and from our observations during field work. As a result, we hereby provide further knowledge on the genus Rhizophora in Thailand that includes a comprehensive update to genus and species descriptions, a key to the species, distributions, habitats and ecology, vernacular names, and uses, in addition to phenological observations, conservation status, etymology, and illustrations for each species. Three names in Rhizophora, are lectotypified: R. apiculata and two associated synonyms of R. mucronata (i.e., R. latifolia Miq. and R. macrorrhiza Griff.). R. longissima Blanco, a synonym of R. mucronata, is considered as a neotype. Besides that, leaf anatomical characters and pollen morphology are presented for completely information of Rhizophora in Thailand.

Materials and Methods

Herbarium specimens deposited in BK, BKF, and those included in the virtual herbaria of AAU (https://www.aubot.dk/search_form.php), BM (https://www.nhm.ac.uk/our-science/collections/botany-collections.html), BR (http://www.botanicalcollections.be), CAL (https://ivh.bsi.gov.in/phanerogams), E (https://data.rbge.org.uk/search/herbarium/), G (http://www.ville-ge.ch/cjb/), JSTOR Global Plants (http://plants.jstor.org), K (including K-W) (http://www.kew.org/herbcat), L (including U) (https://bioportal.naturalis.nl/), P (https://science.mnhn.fr/institution/mnhn/collection/p/item/search), The Wallich Catalogue Online (https://wallich.rbge.org.uk/), US (https://collections.nmnh.si.edu/search/botany/), and W (https://www.nhm-wien.ac.at/en/research/botany) were examined by consulting taxonomic literature (e.g., Kurz, 1877; Henslow, 1878; Trimen, 1894; King, 1897; Guillaumin, 1920; Hou, 1958, 1970; Backer & Bakhuizen van den Brink, 1963; McCusker, 1984; Qin & Boufford, 2007; Duke, 2010; Shamin-Shazwan et al., 2021). All herbaria acronyms follow Thiers (2023).

All specimens cited have been seen by the authors unless stated otherwise. The taxonomic history of the species was compiled using the taxonomic literature and online databases (IPNI, 2023; POWO, 2023). The morphological characters, distributions, habitats and ecology, phenology, and uses were described from historic and newly collected herbarium specimens and the author’s observations during field work. The vernacular names were compiled from the specimens examined and the literature (e.g., Hou, 1970; Pooma & Suddee, 2014). Thailand floristic regions follow Flora of Thailand. Vol. 4(3.3) (The Forest Herbarium, Department of National Parks, Wildlife & Plant Conservation, 2023). The assessment of conservation status was performed following the IUCN Red List Categories and Criteria (IUCN Standards & Petitions Committee, 2022) for a preliminary assessment of the conservation category in combination with GeoCAT analysis (Bachman et al., 2011) and field information. The calculation of Extent of Occurrence (EOO) and Area of Occupancy (AOO) are based on GeoCAT (https://www.kew.org/science/our-science/projects/geocat-geospatial-conservation-assessment-tool).

Plant samples of Rhizophora apiculata and R. mucronata for anatomical study were collected from the International Mangrove Botanical Garden Rama IX, Chanthaburi province. The leaf anatomical characters of these two species were investigated by transverse sectioning with a sliding microtome at 15–20 μm thickness. For the study of epidermal cells of leaves, they were peeled and mounted. The permanent slides of leaves were made following the standard methods of Johansen (1940) and Kermanee (2008). The anatomical characteristics were investigated and recorded photographically with an Olympus BX53 microscope and an Olympus DP74 microscope digital camera at the Department of Botany, Faculty of Science, Kasetsart University (KU). The anatomical terminologies follow those in the study by Metcalfe & Chalk (1957).

The samples of pollen grains were taken from the specimen collected from the International Mangrove Botanical Garden Rama IX, Nong Bua subdistrict, Mueang Chanthaburi district, Chanthaburi province [Rhizophora apiculata (C. Ngernsaengsaruay, P. Chanton & P. Aksornwachpalin IMBG003, IMBG007) and R. mucronata (C. Ngernsaengsaruay, P. Chanton & P. Aksornwachpalin IMBG006)]. They were examined and recorded photographically with an Olympus BX53 microscope and an Olympus DP74 microscope digital camera. Materials were prepared for scanning electron microscopy (SEM) at the Scientific Equipment Centre, Faculty of Science, Kasetsart University by mounting pollen grains on stubs using double-sided sellotape, sputter-coating them with gold and examining them using an FEI Quanta 450 SEM (Hillsboro, OR, USA) at 15.00 KV. The characteristics of pollen grains were examined and measured, following Erdtman (1945, 1952) and Simpson (2010). The pollen morphology terminologies follow those of Punt et al. (2007).

We obtained permission to collect specimens from the International Mangrove Botanical Garden Rama IX, Department of Marine and Coastal Resources, DMCR (ทส) 0426/1471.

Results

Taxonomic treatment

Rhizophora L., Sp. Pl. 1: 443. 1753 et Gen. Pl. ed. 5: 202. 1754; Lam., Encycl. 6: 187. 1804; Blume, Enum. Pl. Javae 1: 91. 1827; DC., Prodr. 3: 31. 1828; Rchb., Deut. Bot. Herb.-Buch: 171. 1841; Blume, Mus. Bot. 1(9): 131. 1849; Miq., Fl. Ned. Ind., Eerste Bijv. [Fl. Ind. Bat.] 1(1): 583. 1855; Kurz, Forest Fl. Burma 1: 447. 1877; G. Hensl. in Hook. f., Fl. Brit. India 2: 435. 1878; Trimen, Handb. Fl. Ceylon 2: 150. 1894; King, J. Asiat. Soc. Bengal, Pt. 2, Nat. Hist. 66(1): 312. 1897; Guillaumin in Lecomte et al., Fl. Indo-Chine 2(6): 721. 1920; Ding Hou in Steenis, Fl. Males., Ser. 1, Spermat. 5(4): 448. 1958 et Blumea 10(2): 625. 1960; Backer & Bakh. f., Fl. Java 1: 379. 1963; S. A. Graham, J. Arnold Arbor. 45(3): 286. 1964; V. C. Vu, Fl. Cambodge Laos Vietnam 4: 136. 1965; Ding Hou in Smitinand & K. Larsen, Fl. Thailand 2(1): 6. 1970; A. McCusker in A. S. George et al., Fl. Australia 22: 1. 1984; H. Qin & Boufford, Fl. China 13: 295. 2007; Schwarzb. in Kubitzki, Fam. & Gen. Vasc. Pl. 11: 293. 2014; Ghafoor, Fl. Pakistan (http://www.tropicos.org/Name/40024808).— Mangle Adans., Fam. Pl. 2: 445. 1763.— Mangium Rumph. ex Scop., Intr. Hist. Nat.: 218. 1777.— Aerope (Endl.) Rchb., Deut. Bot. Herb.-Buch: 171. 1841. Type species: Rhizophora mangle L. (lectotype, designated by N. L. Britton, N. Amer. Trees 716 (6 May 1908), Index Nominum Genericorum (ING), https://naturalhistory2.si.edu/botany/ing/).

Description. Habit evergreen trees. Stilt roots numerous branched, descending from the base of the stem. Branches opposite and decussate, growing upward at acute angles; branchlets with conspicuous annular stipular scars and leaf scars at the nodes. Stipules interpetiolar, in pairs enclosing the young shoot (including the terminal bud and young leaves) and young inflorescences, forming a narrowly conical, gradually narrowing towards the apex, glabrous and caducous, depressed orbicular at the basal part and suborbicular to orbicular at the apical part in outline in transverse section, with dense colleters inside at the base, aggregated as a band, which produce and release a sticky exudate. Leaves opposite and decussate; laminae elliptic, narrowly elliptic, lanceolate-ovate or ovate, apex mucronate by the extending midrib, a short terminal stiff point, usually caducous, margin entire, coriaceous, glabrous on both surfaces, with numerous, scattered tiny black cork warts (tiny black dot-like structures) below; petioled. Inflorescences axillary or leafless branchlets (in the axils of leaf scars), simple dichasium, 2-flowered or compound dichasium, dichotomously branched, 3–9-flowered cymes (5–16-flowered in young inflorescences, some flowers falling off before mature), peduncled. Bracteoles at the base of the flower, completely connate, cup-shaped or connate at the base, bilobed. Flowers bisexual, 4-merous, sessile or pedicelled; mature flower buds broadly ellipsoid, ellipsoid, ovoid or conical-ovoid; hypanthium short; sepals 4, inserted on the rim of the hypanthium, valvate, erect or patent after anthesis, coriaceous or thickly coriaceous; petals 4, inserted at the base of the disk, valvate, alternisepalous, the edges barely enclosing the single antepetalous stamen, ephemeral, caducous, smaller than the sepals, thin, unlobed, without appendages, hairy or glabrous; stamens 2–3 times the number of petals, inserted on the margin of the disk; anthers introrse, multi-locellate (in longitudinal section), dehiscing with an adaxial valve; filaments very short (much shorter than anthers); ovary half-inferior, adnate to the hypanthium, 2-locular, with two ovules per locule; style very short; stigma 2-lobed. Fruits baccate: ovoid, conical-ovoid or broadly ovoid (before seed germination) and obpyriform, pendulous (when the hypocotyls protruding from the fruits); persistent sepals patent in young fruits and then reflexed in mature fruits; pedunculate. Seeds usually 1, viviparous. Hypocotyls pendulous, cylindrical-clavate, perforating and protruding the apex of the mature fruits and ultimately disarticulating at the junction with the cotyledons; cotyledons thick, fused into a cylindrical tube (cotyledonous tube), protruding from the apex of the fruits and withering with it after the seedling has fallen; young shoots (plumules) forming a narrowly conical at the apex of the hypocotyls.

A genus of six species, of which three species in the Indian subcontinent, China and Southeast Asia (Rhizophora apiculata, R. mucronata, and R. stylosa) and two species in Thailand.

Morphologically, Rhizophora is related to Bruguiera, Ceriops, and Kandelia in its mangrove habitat; viviparous germination (seeds germinating inside the fruit while this is still attached to the tree and hypocotyls protruding from the fruits); interpetiolar stipules in pairs enclosing the terminal bud and young leaves, with colleters inside at the base; and decussate, entire, coriaceous and glabrous leaves. Rhizophora can be distinguished by its stems with numerous branched stilt roots; leaves with numerous, scattered tiny black cork warts below; 4-merous flowers; unlobed petals, without appendages; stamens 2–3 times the number of petals; introrse, multi-locellate anthers, dehiscing with an adaxial valve; and half-inferior, 2-locular ovaries, two ovules per locule.

A comparison of the morphological characteristics of these four related genera is shown in Table 1. The characters of Bruguiera, Ceriops, and Kandelia are taken from previous studies (e.g., Kurz, 1877; Henslow, 1878; Trimen, 1894; King, 1897; Merrill, 1912; Guillaumin, 1920; Ridley, 1922; Hou, 1958, 1970; Backer & Bakhuizen van den Brink, 1963; Graham, 1964; McCusker, 1984; Schwarzbach, 2014; Qin & Boufford, 2007; Sheue, 2003; Sheue, Chen & Yang, 2012) and also from the author’s examination of herbarium specimens and observations during the field work.

Table 1 A comparison of the morphological characteristics of Rhizophora and related genera, Bruguiera, Ceriops, and Kandelia in Rhizophoreae.

License: CC0 1.0.

Characteristics	Tribe Rhizophoreae	
Rhizophora	Bruguiera	Ceriops	Kandelia	
Habit and habitats	True mangrove trees	True mangrove trees	True mangrove small to medium-sized trees or shrubs	True mangrove small trees	
Specialized roots	Stems supported by numerous branched stilt roots	Stems with basal buttresses and with knee-like or knobby pneumatophores produced from horizontal roots	Stems with appressed stilt roots	None	
Stipule length (cm)	4.5–12.5	2–6.5	1–3.5	2.5–4	
Stipule shapes in outline in transverse section	Depressed orbicular at the basal part, suborbicular to orbicular at the apical part	Depressed orbicular at the basal part, suborbicular to orbicular at the apical part	Flattened	Flattened	
Aggregated colleter shapes	Band	Rectangular-trapezoid, trapezoid or trapezoid-semicircle	Triangular	Triangular	
Colletter shapes and stalk	Narrowly conical, sessile	Finger-like rod, short stalked	Finger-like rod, long stalked	Finger-like rod, long stalked	
Cork warts on leaf surfaces	With numerous, scattered tiny black cork warts on the abaxial leaf surface	Sometimes may appear on both adaxial and abaxial leaf surfaces	Sometimes may appear on both adaxial and abaxial leaf surfaces	sometimes may appear on both adaxial and abaxial leaf surfaces	
Leaf apex	Mucronate by the extending midrib, a short terminal stiff point usually caducous	Acute	Obtuse, rounded, slightly emarginate to emarginate	Obtuse	
Leaf scars	With many tiny vascular bundle strands, arranged in a crescent moon	With three major vascular bundle strands, similar in size, loosely arrangement	With three major vascular bundle strands, similar in size, closely arrangement	With three major vascular bundle strands, the two upper vascular bundle strands distinctly larger than the lower one	
Inflorescences	Peduncled, simple, 2-flowered or dichotomously branched, 3–9-flowered cymes (5–16-flowered in young inflorescences)	Peduncled, 2–5-flowered cymes or solitary	Subsessile to shortly peduncled, (2–)4–many-flowered condensed cymes	Peduncled, dichotomously branched, 4–9-flowered cymes	
Bracteoles	Completely connate, cup-shaped or connate at the base, bilobed	Ebracteolate	2, partly connate, cup-shaped	2–4, connate, cup-shaped and adnate to the base of the calyx	
Flowers	4-merous	8–16-merous (polymerous)	5–6-merous	5(–6)-merous	
Sepals	4, elliptic-oblong or triangular, erect or patent after anthesis	8–16, subulate-lanceolate, erect, slightly patent or ascending after anthesis	5–6, ovate, erect after anthesis	5(–6), linear-oblong or linear, reflexed after anthesis	
Petals	4, apex unlobed, without appendages; 0.5–1.2 cm long, hairy or glabrous	8–16, apex bilobed or emarginate; 0.2–1.5 cm long, hairy	5–6, apex emarginate or truncate, fringed or with 3 clavate appendages (occasionally only 2); 2.5–4 mm long, hairy or glabrous	5(–6), apex bilobed, with long seta in the sinus, each lobe multifid; 1.4–1.5 cm long, glabrous	
Stamens	(2–)3 times the number of petals [(8–)12(–14)]; anthers multi-locellate (in longitudinal section), dehiscing with an adaxial valve (dehiscing introrsely)	2 times the number of petals (16–32); anthers 4-locular, dehiscing with lengthwise slits	2 times the number of petals (10–12); anthers 4-locular, dehiscing with lengthwise slits	Numerous (30–40); anthers 4-locular, dehiscing with lengthwise slits	
Ovary	Half-inferior, 2-locular; ovules 2 per locule	Inferior, 2–4-locular; ovules 2 per locule	Half-inferior, 3-locular; ovules 2 per locule	Inferior, 1-locular; ovules 6 per locule	
Persistent sepals in mature fruits	Reflexed	Ascending, patent, erect or reflexed	Patent, reflexed, erect or ascending	Reflexed	
Hypocotyls	Terete	Obscurely ridged or terete	Ridged and grooved	Terete	

A key to the species of Rhizophora in Thailand

1a. Leaf laminae elliptic, narrowly elliptic or lanceolate-ovate, widest up to 8.5 cm, a terminal stiff point less than 3 mm long; inflorescences simple, 2-flowered cymes, on leafless branchlets (in the axils of leaf scars); bracteoles brown, connate, cup-shaped, shallowly lobed; flowers sessile; mature flower buds broadly ellipsoid or ellipsoid (widest at the middle); sepals elliptic-oblong, pale green, usually tinged with brown finely reticulated pattern outside, thickly coriaceous; petals oblong, flattened, not involute, glabrous; stamens mostly 12; fruits smaller, 1.5–2.6 cm long, 1–1.4 cm in diam. at the base (before seed germination) and 2.2–3.3 cm long, 1.2–1.8 cm at the base (when the hypocotyls protruding from the fruits); hypocotyls dark green or purplish dark green, glossy, usually smaller; cotyledonous cylindrical tubes red, dark greenish red or reddish dark green, 5–8 mm in diam.; colleters bright yellow, with a sticky clear exudate……..… 1. Rhizophora apiculata 1b. Leaf laminae ovate or lanceolate-ovate, widest up to 13 cm, a terminal stiff point more than 3 mm long; inflorescences dichotomously branched, 3–9-flowered cymes (5–16-flowered in young inflorescences), axillary; bracteoles pale green, connate at the base, bilobed; flowers pedicelled; mature flower buds ovoid, sometimes conical-ovoid (widest at the base); sepals triangular, pale green, greenish pale yellow or pale yellow outside, coriaceous; petals lanceolate-ovate or lanceolate, involute, villous along margins; stamens 8; fruits larger, 3.7–5 cm long, 2.1–2.8 cm in diam. at the base (before seed germination) and 4–7 cm long, 2.4–3.5 cm in diam. at the base (when the hypocotyls protruding from the fruits); hypocotyls green, slightly glossy, usually larger; cotyledonous cylindrical tubes pale green or greenish pale yellow, up to 1 cm in diam.; colleters pale yellow, with a sticky white exudate........................................................................................ 2. Rhizophora mucronata 1. Rhizophora apiculata Blume, Enum. Pl. Javae 1: 91. 1827; Hochr., Candollea 2: 446. 1925; Ding Hou, Fl. Males., Ser. 1, Spermat. 5(4): 452. figs. 7, 11 and 12. 1958 et Blumea 10(2): 630. 1960; Backer & Bakh. f., Fl. Java 1: 379. 1963; V. C. Vu, Fl. Cambodge Laos Vietnam 4: 138. t. 1, figs. 10 and 11. 1965; Ding Hou in Smitinand & K. Larsen, Fl. Thailand 2(1): 6. 1970; Toml., J. Arnold Arbor. 59: 163. 1978; N. C. Duke & J. S. Bunt, Aust. J. Bot. 27: 667. 1979; A. McCusker in A. S. George et al., Fl. Australia 22: 3. 1984; H. Qin & Boufford, Fl. China 13: 295. 2007; N. C. Duke, Blumea 55: 177. fig. 5. 2010; Shamin-Shazwan et al., Science Heritage J. 5(1): 2. fig. 1. 2021; Ghafoor, Fl. Pakistan (http://www.tropicos.org/Name/27600231). Type: Indonesia, Java, vivipary, s.d., Unknown s.n. (Herb. Lugd. Batav. 908.191–276) (lectotype, designated here L [L0009917, photo seen] (Fig. 1); isolectotype L [L0009918, photo seen]).

Figure 1 Lectotype of Rhizophora apiculata, Unknown s.n. (Herb. Lugd. Batav. 908.191-276) (L [L0009917]) from Java, Indonesia, designated here. Photo: Naturalis Biodiversity Center, Leiden, the Netherlands, https://bioportal.naturalis.nl/en/specimen/L__0009917.

= Rhizophora candelaria DC., Prodr. 3: 32. 1828; Trimen, Handb. Fl. Ceylon 2: 151. 1894; Merr., Philipp. J. Sci., C 9: 118. 1914; Merr., Enum. Philipp. Fl. Pl. 3: 145. 1923; Craib, Fl. Siam. 1(4): 592. 1931. Type: icon, Pee-Kandel Rheede, Hort. Malab. 6: 61. t. 34 (van Rheede, 1686) (lectotype, designated by Merrill, 1914: 118).

= Rhizophora conjugata [auct. non L.] Arn., Ann. Mag. Nat. Hist. 1(5): 363. 1838; Blume, Mus. Bot. 1(9): 134. 1849; Miq., Fl. Ned. Ind., Eerste Bijv. [Fl. Ind. Bat.] 1(1): 584. 1855; Kurz, Forest Fl. Burma 1: 447. 1877; G. Hensl. in Hook. f., Fl. Brit. India 2: 436. 1878; King, J. Asiat. Soc. Bengal, Pt. 2, Nat. Hist. 66(1): 313. 1897; Merr., Philipp. J. Sci. 1. Suppl. 1: 102. 1906 et Fl. Manila: 347. 1912; Guillaumin in Lecomte et al., Fl. Indo-Chine 2(6): 724. 1920; Ridl., Fl. Malay Penins. 1: 693. 1922.

Description. Habit trees, 2–20(–30) m tall, up to 160 cm girth. Stilt roots extending 1–5 m up the stem. Branches usually with pendulous aerial roots; branchlets terete, glabrous. Bark greyish brown, shallowly fissured; inner bark red. Stipules red, pale greenish red or pale yellowish red linear-lanceolate, 4.5–10 cm long, 0.3–1 cm in diam. at the base (before the leaves emerging), apex acute; colleters sessile, bright yellow, narrowly conical, 0.3–1 mm long, 0.1–0.2 mm in diam. at the base, apex obtuse, with a sticky clear exudate. Leaves: laminae elliptic, narrowly elliptic or lanceolate-ovate, 12–20.5 × 5–8.5 cm, apex mucronate, a short terminal stiff point red when young, pale green when mature, turning black before come off, 1.2–2.8 mm long, base cuneate or obliquely cuneate, margin entire, coriaceous, shiny dark green above, pale green below, glabrous on both surfaces, with numerous, scattered tiny black cork warts below, midrib usually pale green (paler than laminae), the uppermost pair of midribs red, flattened above, raised below, secondary veins 8–14 pairs, curving towards the margin connected in distinct loops and united into an intramarginal vein, visible on both surfaces, with intersecondary veins, veinlets reticulate, visible above, obscure below; petioles usually green, the uppermost pair of petioles red or green, 1.3–4 cm long, 3–5 mm in diam., somewhat flattened, glabrous; young leaves shiny pale green; mature leaves turning greenish bright yellow and bright yellow before falling off; dry leaves yellowish brown. Inflorescences on leafless branchlets (in the axils of leaf scars), simple dichasium, 2-flowered cymes, 1.6–3 cm long; peduncles green, 0.4–1.4 cm long, 3.5–8.5 mm in diam., somewhat flattened, thick, glabrous. Bracteoles at the base of the flower, brown, connate, cup-shaped, shallowly lobed, fleshy. Flowers sessile; mature flower buds broadly ellipsoid or ellipsoid, 0.9–1.6 cm long, 0.7–1.2 cm in diam., apex acute; opened flowers 1.3–2.2 cm in diam.; sepals 4, erect or patent after anthesis, pale green, usually tinged with brown finely reticulated pattern outside, yellow tinged with pale green at the base inside, concave, elliptic-oblong, 0.9–1.3 × 0.4–0.7 cm, apex acute, thickly coriaceous, glabrous; petals 4, creamish white, narrowly oblong or linear-narrowly oblong, 0.5–1.2 × 0.1–0.2 cm, thin, curved upwards in flower buds, flattened after anthesis, not involute, glabrous; stamens (8–)12(–14) (in case of 12: 4 pairs antesepalous and 4 antepetalous); anthers falcate-like, 5.8–9.5 × 1.3–3 mm, apex acute, triangular in outline in transverse section; filaments very short; free part of the ovary 2–3.4 mm long, 1.6–3.7 mm in diam., with 7–10 radiate ridges; style very short; stigma 2-lobed. Fruits greenish brown: ovoid, sometimes broadly ovoid, 1.5–2.6 cm long, 1–1.4 cm in diam. at the base, apex rounded (before seed germination) and obpyriform, 2.2–3.3 cm long, 1.2–1.8 cm in diam. at the base, 0.8–1.2 cm in diam. at the apex (when the hypocotyls nearly come off), roughened surface, scattered tuberculate; persistent sepals reflexed, elliptic-oblong, 1.1–1.9 × 0.6–0.8 cm; infructescence stalks 0.6–2.3 cm long, 3.5–9 mm in diam., without fruit stalks, with fleshy bracteoles at the base of the fruit. Seeds 1, viviparous. Hypocotyls dark green or purplish dark green, glossy, cylindrical-clavate, 25–45 cm long, medullary 0.6–1.1 cm in diam., basally 0.8–1.3 cm in diam., acute at the basal end, with slight longitudinal ridges, roughened surface, with numerous, scattered lenticels; cotyledonous cylindrical tubes red, dark greenish red or reddish dark green, 5–8 mm in diam. (can be seen when the hypocotyls nearly falling off) (Figs. 2 and 3).

Figure 2 Rhizophora apiculata. (A) Flowering branchlets showing leaves and inflorescences with flower buds and an opened flower. (B and C) Flowering branchlets. (D) 2-flowered cyme with cup-shaped bracteole at the base of the flower buds. (E) An opened flower. (F) Sepal (inside). (G) Stamens. (H) Pistil and hypanthium. (I) Flowering and fruiting branchlets. (J) Fruit with persistent sepals and viviparous seed (seedling) showing a cylindrical-clavate hypocotyl. Photo: Drawn by Wanwisa Bhuchaisri.

Figure 3 Rhizophora apiculata. (A) Habitat and habit. (B) Stems supported by numerous branched stilt roots. (C) Branches with pendulous aerial roots. (D) Stem, outer bark and inner bark. (E) Branchlets and leaves. (F) Mucronate leaf apex and leaf lamina with numerous, scattered tiny black cork warts on the abaxial surface. (G) Terminal interpetiolar stipule with dense colleters inside at the base and branchlets with leaf scars. (H) Leaf scar with many tiny vascular bundle strands, arranged in a crescent moon. (I) Petiole with many tiny vascular bundle strands, arranged in a crescent moon. (J and K) Flowering branchlets showing inflorescences with flower buds and an opened flower. (L) Flower bud in longitudinal section. (M) Flower bud in transverse section. (N and O) Flowering and fruiting branchlets. (P) Mature fruits with persistent sepals. (Q–T) Flowering and fruiting branchlets showing successive stages in development of hypocotyl. (U and V) Fruits showing successive stages in development of hypocotyl. (W and X) Fruits with seedling, hypocotyl disarticulated from cotyledonous cylindrical tube, revealing plumule. Photos: Chatchai Ngernsaengsaruay (A–K and N–X); Pichet Chanton (L and M).

Distribution. Pakistan, India (Odisha, Andhra Pradesh, Tamil Nadu, Karnataka, Andaman and Nicobar Islands), Bangladesh [Chittagong], Sri Lanka, Myanmar [Arakan, Bago (formerly spelled Pegu), Mergui Archipelago], China [South Guangxi, Hainan], Vietnam, Cambodia, Thailand, Peninsular Malaysia (also called Malaya) [Kedah, Langkawi (West of Kedah), Pulau Pinang, Perak, Selangor, Negeri Sembilan (also called Negri Sembilan), Malacca (also called Melaka), Johor (also called Johore)], Singapore, Indonesia [Sumatra, Bangka, Java, Lesser Sunda Islands, Sulawesi (also called Celebes), Moluccas (also called Maluku), New Guinea (Papua)], Borneo [Malaysia (Sarawak, Sabah), Brunei, Indonesia (Kalimantan)], Philippines [Luzon, Marinduque, Oriental Mindoro, Sibuyan Island, Cuyo Archipelago, Cebu, Eastern Samar, Zamboanga, Basilan, Mindanao, Sulu Archipelago, Palawan], East Timor (also called Timor-Leste), Australia (Northern Territory, Queensland, Cocos (Keeling) Islands), Micronesia [Marianas Islands, Palau Islands, Caroline Islands], Melanesia [Bismarck Archipelago, Papua New Guinea, Solomon Islands, Vanuatu, New Caledonia].

Distribution in Thailand. South-Western: Phetchaburi, Prachuap Khiri Khan; Central: Bangkok, Samut Prakan, Samut Songkhram, Samut Sakhon; South-Eastern: Chachoengsao, Chon Buri, Rayong, Chanthaburi, Trat; Peninsular: Chumphon, Ranong, Surat Thani, Phangnga, Phuket, Krabi, Nakhon Si Thammarat, Phatthalung, Trang, Satun, Songkhla, Pattani, Narathiwat.

Habitat and Ecology. Rhizophora apiculata is restricted to mangrove forests, gregarious on the deep soft mud of estuaries, especially along riverbanks or creeks flooded by high tides, where it is often the dominant species. Rhizophora species (R. apiculata and R. mucronata) constitute the typical element of this vegetation type, but where other genera co-occur, there is often a distinct zonation with Ceriops, Bruguiera, and Kandelia. Rhizophora commonly dominates the seaward fringe of the true mangrove species, sometimes associated with Avicennia (Lamiaceae) and Sonneratia (Lythraceae). It is most frequently found mixed with R. mucronata.

Phenology. Flowering, fruiting and viviparous germination more than once, nearly throughout the year.

Conservation status. Least Concern (LC) (Duke et al., 2010). This species is very widely distributed from Africa, China to Tropical Asia, Australia to Micronesia and Melanesia, and has a large extent of occurrence (EOO of 187,200,667.98 km2) and a relatively large area of occupancy (AOO of 880 km2). Because of this wide distribution and the number of localities, it is appropriate to consider its status as LC.

Etymology. The specific epithet of Rhizophora apiculata is a Latin word meaning ending abruptly in a short point (Stearn, 1992; Gledhill, 2002) and refers to an apiculate leaf apex formed by extending midrib. However, we suggest using mucronate, ending abruptly in a short stiff point, but shorter than those of R. mucronata.

Vernacular name. Kongkang (โกงกาง) (Ranong); Kongkang bai lek (โกงกางใบเล็ก) (Central); Phangka sai (พังกาทราย) (Krabi); Phangka bai lek (พังกาใบเล็ก) (Phangnga); Tall-stilt mangrove (English).

Uses. The wood is used for firewood and for making charcoal. Wood poles are used for piling and construction purposes. The timber is used for making furniture. The bark contains tannin, used for tanning leather and used for dyeing, toughening fishing nets, lines and ropes. In Ban Yisan, Yisan subdistrict, Amphawa district, Samut Songkhram province, Rhizophora apiculata is usually planted for its wood and is used for making charcoal (Fig. 4).

Figure 4 Uses of Rhizophora apiculata in Thailand. (A–D) In Ban Yisan, Yisan subdistrict, Amphawa district, Samut Songkhram province, the wood is used for making charcoal. Photos: Chatchai Ngernsaengsaruay.

Lectotypification. Rhizophora apiculata was named by Blume (1827), based on the specimen collected from Java but he did not mention the collector number and the name of the herbarium where the materials were kept. The lectotype of R. apiculata had already been annotated as such presumably by Ding Hou, although apparently not published (L L0009917 and L0009918). We located two sheets of the specimen Unknown s.n. collected from Java (Herb. Lugd. Batav. 908.191-276) at L [L0009917 and L0009918] and following Art. 9.6 of the ICN (Turland et al., 2018), they constitute syntypes. Therefore, the L [L0009917] specimen is in the best condition and clearly shows the diagnostic characters for the species and is selected here as the lectotype, following Art. 9.3 and 9.12 of the ICN (Turland et al., 2018).

Notes. Rhizophora conjugata sensu Arnott (1838: 363) is a nom. illeg., but in the protologue Arnott includes the name and type of R. candelaria DC. It is therefore a nomenclatural, homotypic synonym of R. candelaria.

The vernacular name “Kongkang bai lek” meaning small-leaved kongkang or small-leaved Rhizophora. The leaves of this species are usually smaller than Kongkang bai yai (Rhizophora mucronata).

Rhizophora apiculata is allied to R. × lamarckii. It can be separated from the latter by the glabrous petals (vs the petals with short hairs along the margins and sometimes also on the inside); the inflorescences on leafless branchlets (in the axils of leaf scars), usually 2-flowered (vs the inflorescences mostly in the axils of leaves, 2(–4)-flowered); and the sessile flowers (vs the usually short-pedicelled flowers). The characters of R. × lamarckii were taken from Hou, 1960. R. × lamarckii is distributed from Tropical Asia to South-West. Pacific, North Northern Territories to North and North-East Queensland (POWO, 2023).

Additional specimens examined. Thailand. South-Western: Phetchaburi [Hat Chao Samran subdistr., 26 May 2007, C. Ngernsaengsaruay own observation]; Prachuap Khiri Khan [Khao Ta Mong Lai, Mueang Prachuap Khiri Khan, fl., 22 Oct 1944, T. Sindhiphong s.n. (BKF [SN049158]); Hua Hin, fl., 18 Jul 1952, T. Smitinand 1465 (BKF [SN049159]); Khlong Khao Daeng, Khao Sam Roi Yot National Park, Kui Buri distr., 28 May 2007, C. Ngernsaengsaruay own observation; Pran Buri River, Pran Buri distr., 28 May 2007, C. Ngernsaengsaruay own observation; Khlong Wan subdistr., 10 Jun 2007, C. Ngernsaengsaruay own observation]; Central: Bangkok [Bang Khun Thian, 23 Jun 2019, C. Ngernsaengsaruay own observation; Bang Khun Thian, reported by Consultants of Technology Co., Ltd. & Panya Consultants Co., Ltd. (2007)]; Samut Prakan [En route from Ban Bang Pu Kao to Samut Prakan province, fl., 14 Aug 1967, T. Shimizu, H. Koyama & N. Fukuoka T-7582 (BKF [SN049157]); Ban Khlong Dan, fl., 14 Jun 1968, H. M. van der Kevie 8 (L [L2496451]); Bang Phli distr., fl., vivipary, 23 Aug 1972, S. Sahpahnuchat 18 (BK); Bang Kachao, Song Khanong subdistr., Phra Pradaeng distr., 30 Mar 2012, C. Ngernsaengsaruay own observation]; Samut Songkhram [Mueang distr., vivipary, 23 Aug 1975, J. F. Maxwell 75-915 (BK, L [L2496407]); Khlong Khon distr., 18 Jul 2016, C. Ngernsaengsaruay own observation; Ban Yisan, Yisan subdistr., Amphawa distr., 31 December 2023, C. Ngernsaengsaruay own observation]; Samut Sakhon [Ao Mahachai, 19 Jul 2016, C. Ngernsaengsaruay own observation]; South-Eastern: Chachoengsao [Bang Pakong distr., 18 Jan 2019, C. Ngernsaengsaruay own observation; Bang Pakong distr., reported by Lumsun, Kutintara & Emphandhu (2013)]; Chon Buri [Si Racha distr., vivipary, 28 Aug 1924 [as Rhizophora candelaria], D. J. Collins 1029 (BK, US [US03044507]); Wat Laem Chabang, Saen Suk subdistr., Si Racha distr., fl., 15 Dec 2007, J. F. Maxwell 07-722 (L [L4207203])]; Rayong [Ban Phe, vivipary, 16 Feb 1980, U. Treesukhon s.n. (BKF [SN049189]); Ko Ku Dee, fl., 12 Dec 1988, C. Niyomdham 1959 (BKF [SN049167]); Between Phe to Wang Kaeo, fl., March 1989, C. Phengklai et al. 6821 (BKF [SN049182, SN199626])]; Chanthaburi [Wan Chao, fl., Nov 1924, [as Rhizophora candelaria], Unreadable s.n. (BK SN214348); Khlung distr., vivipary, 26 Jul 1975, T. Smitinand 12071 (BKF [SN049150]); Locality not specified, vivipary, 16 May 1980, K. Iwatsuki & T. Santisuk s.n. (BKF [SN049165]); Khlung distr., fl., 9 Dec 1983, N. Fukuoka & M. Ito T-34908 (BKF [SN099911], L [L4198576]); Khlung distr., fl., 25 Sep 1984, N. Fukuoka T-36274 (BKF [SN107754]); the International Mangrove Botanical Garden Rama IX, Nong Bua subdistr., Mueang Chanthaburi distr., fl., 23 Sep 2023, C. Ngernsaengsaruay, P. Chanton & P. Aksornwachpalin IMBG001, IMBG002, IMBG003; ibid. fl., fr., vivipary, 24 Nov 2023, C. Ngernsaengsaruay, P. Chanton & P. Aksornwachpalin IMBG007]; Trat [Ko Chang, fl., 5 Apr 1923 [as Rhizophora candelaria], Unreadable s.n. (BK [SN214349]); Khlong Mayom, Ko Chang, fl., 2 Jul 1955, B. Sangkhachand 468 (BKF [SN049184]); Locality not specified, fl., 19 Jan 1958, T. Sørensen, K. Larsen & B. Hansen 535 (BKF [SN049147], L [L2496437]); Khlong Mayom, Ko Chang, fl., 2 Apr 1959, T. Sørensen, K. Larsen & B. Hansen 7107 (L [L2496438]); Ko Chang, West side, fl., 18 Nov 1970, C. Charoenphol, K. Larsen & E. Warncke 4986 (L [L2496449], P [P05552015]); Ko Chang, fl., 4 Aug 1973, G. Murata, N. Fukuoka & C. Phengklai T-17456 (BKF [SN049155], L [L2496409]); Ko Kut, fl., 20 Oct 2000, C. Phengklai et al. 13090 (BKF [SN130495]); Klong Chao, Ko Kut, fl., vivipary, 6 Apr 2002, C. Phengklai et al. 13377 (BKF [SN135602, SN135603, SN135650]); Ao Salat, Ko Kut, fl., 9 Apr 2002, C. Phengklai et al. 13558 (BKF [SN137075]); Ko Kut, fl., 31 Dec 2006, C. Phengklai et al. 15498 (BKF [SN163825]); Ban Salak Khok, Ko Chang, 10 Jul 2022, H. Balslev et al. 10202 (AAU); Ko Kut, 6 Jul 2012, C. Ngernsaengsaruay own observation]; Peninsular: Chumphon [Tako, Lang Suan distr., fl., 18 Jun 1928 [as Rhizophora candelaria], P. Praisurin (Put) 1714 (BK); Khlong Tako, Thung Tako distr., fl., 14 Sep 1934, Smanwanakit 62 (BKF [SN049185, SN199635]); Pathio distr., fl., 12 Aug 2011, C. Phengklai et al. 16162 (BKF [SN192959]) BanI Let, Hat Sai Ri subdistr., Mueang Chumphon distr., 14 May 2008, C. Ngernsaengsaruay own observation; Mu Ko Chumphon National Park, Hat Sai Ri subdistr., Mueang Chumphon distr., 14 May 2008, C. Ngernsaengsaruay own observation]; Ranong [2 kms West of Ranong, vivipary, 16 Jul 1963, King et al. 5574 (L [L2496436], US [US03044503]); Locality not specified, vivipary, 5 May 1968, Kapoe distr., fl., 25 Dec 1976, T. Smitinand 853 (BKF [SN174569, SN210686, SN210687]); C. F. van Beusekom & C. Phengklai 592 (E [E01058154], L [L2496411], P [P05552026]); Kapoe distr., fl., 31 Oct 1981, Pipat s.n. (BKF [SN049172]); Kapoe distr., fl., 25 Dec 1983, N. Fukuoka & M. Ito T-35537 (BKF [SN049168], L [L4198580]); Ban Sam Nak, Kapoe distr., fl., 24 Dec 1983, N. Fukuoka & M. Ito T-35516 (BKF [SN049169]); Khlong Kamphuan, Andaman Coastal Research Station for Development, Kamphuan subdistr., Suk Samran distr., 10 Jul 2009, C. Ngernsaengsaruay own observation]; Surat Thani [Ko Tao, fl., 20 Mar 1927 [as Rhizophora candelaria], A. F. G. Kerr 12976 (BK); Ko Pha Ngan, fl., 7 Sep 1973, C. Phromdet 19 (E [E01058152], L [L2496413], P [P05552014]); Chaiya distr., vivipary, 11 May 1932, C. Bunchu 62 (BKF [SN049173]); Locality not specified, fl., 31 Dec 1983 [as Rhizophora mucronata], N. Fukuoka & M. Ito T-35693 (BKF [SN099932]); Locality not specified, fl., 31 Dec 1983, N. Fukuoka & M. Ito T-35697 (BKF [SN049166]) Kadae subdistr., Kanchanadit distr., 18 May 2008, C. Ngernsaengsaruay own observation; Tapi River, Khlong Chanak subdistr., Mueang Surat Thani distr., 18 May 2008, C. Ngernsaengsaruay own observation; Phum Riang subdistr., Chaiya distr., 19 May 2008, C. Ngernsaengsaruay own observation; Ko Samui, 17 May 2008, C. Ngernsaengsaruay own observation; Ko Tan, 17 May 2008, C. Ngernsaengsaruay own observation]; Phangnga [Thong Lang Village, Takua Thung distr., fl., 29 Jan 1966, E. Hennipman 3753 (BKF [SN049164], L [L2496435]); Ko Phu, fl., 14 Feb 1966, B. Hansen & T. Smitinand T12339 (L [L2496410]); Takua Thung distr., sterile, 25 Aug 1967 [as Rhizophora mucronata], T. Shimizu, N. Fukuoka, & A. Nalampoon T-8036 (BKF [SN049170]); Phangnga Bay, fl., 1 Sep 1984, N. Fukuoka T-35800 (BKF [SN092505], L [L2496448]); Ko Phra Thong, Khura Buri distr., fl., 5 Apr 2003, C. Phengklai et al. 13777 (BKF [SN141677]); Ko Kho Khao, Takua Pa distr., 19 Jun 2009, C. Ngernsaengsaruay own observation; Ko Panyi subdistr., Mueang Phangnga distr., 19 Jun 2009, C. Ngernsaengsaruay own observation]; Phuket [Ko Phu (originally “Phangnga” on the label), fl., 14 Feb 1966, B. Hansen & T. Smitinand 12339 (BKF [SN049187], L [L2496452]); Ko Mak, sterile, 18 Feb 1966, B. Hansen & T. Smitinand 12364 (BKF [SN049183], P [P05552013])]; Krabi [Mueang Krabi distr., fl., 5 Sep 1982, F. Konta, T. Wongprasert & B. Sangkhachand T-29135 (BKF [SN113635]); Khlong Thom Tai subdistr., Khlong Thom distr., 20 Jun 2009, C. Ngernsaengsaruay own observation; Klong Prasong subdistr., Mueang Krabi distr., 21 Jun 2009, C. Ngernsaengsaruay own observation]; Nakhon Si Thammarat [Locality not specified, fl., 27 May 1995, W. Nanakorn et al. 3596 (BKF [SN233237]; Khlong Khanom Estuary, Khanom distr., 23 Jul 2010, C. Ngernsaengsaruay own observation; Pak Phanang Estuary, Pak Phanang distr., 24 Jul 2010, C. Ngernsaengsaruay own observation; Laem Talumphuk subdistr., Pak Phanang distr., 24 Jul 2010, C. Ngernsaengsaruay own observation]; Phatthalung [reported by Information Center of Marine and Coastal Resources (2018); Pak Phayun distr., 8 Mar 2022, C. Ngernsaengsaruay own observation]; Trang [Ban Yong Sata, fl., Sep 1924 [as Rhizophora conjugata], Unreadable s.n. (BK SN214350); Kantang distr., fl., 20 Sep 1933, P. Praisurin (Put) 295 (BKF [SN049171]); Sikao distr., fl., 27 Oct 1965, C. Boonnab 98 (BKF [SN049186]); Ban Bang Khang Khao, Sikao distr., fl., 4 Apr 2004, S. Gardner & P. Sidisunthorn ST392 (L [L4198359]); Sikao, cultivated in Rajamangala University of Technology Srivijaya, Trang Campus, fl., vivipary, 13 Feb 2018, J. Sittichoke 10 (BKF [SN215684])]; Satun [Locality not specified, fl., 24 Jan 1928 [as Rhizophora candelaria], A. F. G. Kerr 14248 (BK); Khlong Che Samat, fl., 12 Feb 1941, T. Premrasmi s.n. (BKF [SN049188]); Locality not specified, vivipary, 15 Sep 1984, N. Fukuoka T-36140 (BKF [SN049160]); South of Satun, fl., 5 Nov 1990, K. Larsen et al. 41167 (AAU, BKF [SN181557]); Ban Hat Sai Khao, Tan Yong Po subdistr., Mueang Satun distr., 25 Oct 2010, C. Ngernsaengsaruay own observation; Ban Lom Puen, La-ngu distr., 26 Oct 2010, C. Ngernsaengsaruay own observation; Tam Malang subdistr., Mueang Satun distr., fl., 14 Jan 2016, K. Kertsawang 3731 (BKF [SN239545])]; Songkhla [Pak Ro subdistr., Singhanakhon distr., 23 Oct 2010, C. Ngernsaengsaruay own observation; Tha Sa-an, Pha Wong subdistr., Mueang Songkhla distr., 23 Oct 2010, C. Ngernsaengsaruay own observation]; Pattani [reported by Chantrapornsyl (2007); Nong Chik distr., 4 Jun 2021, C. Ngernsaengsaruay own observation]; Narathiwat [reported by Information Center of Marine and Coastal Resources (2018); Mueang Narathiwat distr., 3 Jun 2021, C. Ngernsaengsaruay own observation]; Province not specified, fl., 22 Sep 1911, A. F. G. Kerr 2093 (L [L2496439], P [P05575994], TCD [TCD0016669]).

2. Rhizophora mucronata Poir. in Lam., Encycl. 6: 189. 1804 et Tabl. Encycl. 2, 5(2): 517. t. 396, fig. 2. 1819; DC., Prodr. 3: 32. 1828; Arn., Ann. Mag. Nat. Hist. 1: 362. 1838; Wight, Icon. Pl. Ind. Orient. 1. t. 238. 1839; Blume, Mus. Bot. 1(9): 132. 1849; Miq., Fl. Ned. Ind., Eerste Bijv. [Fl. Ind. Bat.] 1(1): 583. 1855; Kurz, Forest Fl. Burma 1: 447. 1877; G. Hensl. in Hook. f., Fl. Brit. India 2: 435. 1878; Trimen, Handb. Fl. Ceylon 2: 151. 1894; King, J. Asiat. Soc. Bengal, Pt. 2, Nat. Hist. 66(1): 312. 1897; Merr., Fl. Manila: 347. 1912; Guillaumin in Lecomte et al., Fl. Indo-Chine 2(6): 722. fig. 75: 1–4. 1920; Ridl., Fl. Malay Penins. 1: 693. 1922; Merr., Enum. Philipp. Fl. Pl. 3: 145. 1923; Craib, Fl. Siam. 1(4): 592. 1931; Ding Hou, Fl. Males., Ser. 1, Spermat. 5(4): 453. figs. 8G–8J, 9, 10. 1958 et Blumea 10(2): 629. 1960; Backer & Bakh. f., Fl. Java 1: 380. 1963; V. C. Vu, Fl. Cambodge Laos Vietnam 4: 142. t. 1, figs. 6–9. 1965; Ding Hou in Smitinand & K. Larsen, Fl. Thailand 2(1): 7. 1970; Toml., J. Arnold Arbor. 59: 163. 1978; N. C. Duke & J. S. Bunt, Aust. J. Bot. 27: 673. 1979; A. McCusker in A. S. George et al., Fl. Australia 22: 2. 1984; H. Qin & Boufford, Fl. China 13: 296. fig. 316.10. 2007; Shamin-Shazwan et al., Science Heritage J. 5(1): 2. fig. 2. 2021; Fl. Pakistan fig. M. Rafiq (http://www.tropicos.org/Name/27600050). Type: Île-de-France (Island of France, Mauritius), s.d., Unknown s.n. (in the Lamarck herbarium, P-LA, not seen).

≡ Rhizophora mucronata Poir. var. typica A. Schimp., Indo-Malay. Strandfl.: 92. 1891, nom. inval.

= Rhizophora macrorrhiza Griff., Trans. Med. Phys. Soc. Calcutta 8(1): 8. 1836. Type: Myanmar, Mergui, sterile, s.d., W. Griffith s.n. (lectotype, designated here G [G00441352, photo seen]) (Fig. 5).

Figure 5 Lectotype of Rhizophora macrorrhiza, a synonym of R. mucronata, W. Griffith s.n. (G [G00441352]) from Mergui, Myanmar, designated here. Photo: © Conservatoire et Jardin botaniques de la Ville de Genève, Switzerland, https://www.ville-ge.ch/musinfo/bd/cjb/chg/adetail.php?id=313491&lang=fr.

= Rhizophora longissima Blanco, Fl. Filip.: 398. 1837; Merr., Sp. Blancoan.: 283. 1918. Type: Philippines, Palawan, Taytay, fl., May 1913, E. D. Merrill Species Blancoanae 409 (neotype, designated here US [US00623705, photo seen]) (Fig. 6).

Figure 6 Neotype of Rhizophora longissima, a synonym of R. mucronata, E. D. Merrill Species Blancoanae 409 (US [US00623705]) from Taytay, Palawan, Philippines, designated here. Photo: National Museum of Natural History (NMNH), Smithsonian Institution, Washington, District of Columbia (D.C.), U.S.A., http://n2t.net/ark:/65665/39525b4b5-1dc4-41d9-8c04-750187cb58d5.

= Rhizophora latifolia Miq., Fl. Ned. Ind., Eerste Bijv. [Fl. Ind. Bat.] Suppl. 1: 324. 1860. Type: Indonesia, Sumatra, Tapanuli, s.d., J. E. Teijsmann HB1064 (lectotype, designated here L [U0005822, vivipary, photo seen] (Fig. 7); isolectotype L [L0009920, sterile, photo seen]).

Figure 7 Lectotype of Rhizophora latifolia, a synonym of R. mucronata, J. E. Teijsmann HB1064 (L [U0005822]) from Tapanuli, Sumatra, Indonesia, designated here. Photo: Naturalis Biodiversity Center, Leiden, the Netherlands, https://bioportal.naturalis.nl/en/multimedia/U%20%200005822_0944926773. License: CC0 1.0.

= Rhizophora mucronata Poir. f. reducta Hochr., Candollea 2: 446. 1925. Type: Australia, Port Hedland, 6 Feb 1905, B. P. G. Hochreutiner 2867 (lectotype, designated by McCusker (1984), not seen).

Description. Habit trees, 2–20(–30) m tall, up to 160 cm girth. Stilt roots extending 1–5 m up the stem. Branches often with pendulous aerial roots; branchlets terete, glabrous. Bark reddish brown, shallowly fissured; inner bark red. Stipules pale green, reddish pale green or pale greenish red when young, turning pale yellow or reddish pale yellow before falling off, linear-lanceolate, 4.5–12.5 cm long, 0.5–1.7 cm in diam. at the base (before the leaves emerging), apex acute; colleters sessile, pale yellow, narrowly conical, 0.5–2.2 mm long, apex obtuse, with a sticky white exudate. Leaves: laminae ovate or lanceolate-ovate, 13–23.5 × 6–13 cm, apex mucronate, a short terminal stiff point pale green, turning black before come off, 3.2–7 mm long, base cuneate or obliquely cuneate, margin entire, coriaceous, shiny dark green above, pale green below, glabrous on both surfaces, with numerous conspicuous, scattered tiny black cork warts below, midrib pale green (paler than laminae), flattened above, raised below, secondary veins 9–14 pairs, curving towards the margin connected in distinct loops and united into an intramarginal vein, visible on both surfaces, with intersecondary veins, veinlets reticulate, visible above, obscure below; petioles green, 1.7–4.6 cm long, 3–6 mm in diam., slightly flattened, glabrous; young leaves shiny pale green; mature leaves turning greenish bright yellow and bright yellow before falling off; dry leaves yellowish brown. Inflorescences axillary, compound dichasium, dichotomously branched, 3–9-flowered cymes (5–16-flowered in young inflorescences, some flowers falling off before mature), 4–11.2 cm long; peduncles pale green, 2.3–7 cm long, 2–4.5 mm in diam., somewhat flattened, glabrous. Bracteoles at the base of the flower, pale green, connate at the base, bilobed. Flowers: pedicels pale green, 0.6–1.5 cm long, 2.5–4 mm in diam., somewhat flattened; mature flower buds ovoid, sometimes conical-ovoid, 1.1–1.7 cm long, 0.6–1 cm in diam., apex obtuse; opened flowers 1.4–2 cm in diam.; sepals 4, erect or patent after anthesis, pale green, greenish pale yellow or pale yellow outside, pale yellow inside, concave, triangular, 0.9–1.2 × 0.5–0.7 cm, apex acute, coriaceous, glabrous; petals 4, creamish white, lanceolate-ovate or lanceolate, 5–8 × 2–3 mm, thin, involute, villous along margins; stamens 8, 4 antesepalous and 4 antepetalous (each petal enclosing 1 stamen in flower buds); anthers falcate-like, 5.5–7.8 × 1.4–2.2 mm, apex mucronate, triangular in outline in transverse section; filaments very short; free part of the ovary 1.5–3 mm long, 1.8–3.6 mm in diam., with 7–9 radiate ridges; style very short; stigma 2-lobed. Fruits brownish green: ovoid or conical-ovoid, 3.7–5 cm long, 2.1–2.8 cm in diam. at the base, apex obtuse (before seed germination); obpyriform, 4–7 cm long, 2.4–3.5 cm in diam. at the base, 1.1–1.9 cm in diam. at the apex (when the hypocotyls nearly come off), roughened surface, basally often tuberculate, with a sticky white exudate when cut, turning creamish white; persistent sepals reflexed, triangular, 0.9–1.4 × 0.5–0.9 cm; infructescence stalks 2.4–6 cm long, 2.5–4.5 mm in diam.; fruit stalks 0.5–1.3 cm long, 3.5–6 mm in diam. Seeds 1, viviparous. Hypocotyls green, slightly glossy, cylindrical-clavate, 25–70 cm long, medullary 0.9–1.6 cm in diam., basally 0.7–1.7 cm in diam., acute at the basal end, with longitudinal ridges, roughened surface, with numerous, scattered lenticels; cotyledonous cylindrical tubes pale green or greenish pale yellow, 0.7–1 cm in diam. (can be seen when the hypocotyls nearly falling off) (Figs. 8 and 9).

Figure 8 Rhizophora mucronata. (A) Flowering branchlets showing leaves and inflorescences with young and mature flower buds. (B) Branchlet and terminal interpetiolar stipule. (C) Dichotomously branched, several-flowered cyme with mature flower buds and opened flowers. (D) An opened flower. (E) Fruit with persistent sepals. (F) Fruit with persistent sepals and viviparous seed (young seedling) showing a young hypocotyl perforating and protruding the apex of the mature fruits. (G) Fruit with persistent sepals and viviparous seed (longitudinal section). (H) Fruit with persistent sepals and viviparous seed showing a cylindrical-clavate hypocotyl. Photo: Drawn by Wanwisa Bhuchaisri.

Figure 9 Rhizophora mucronata. (A) Habitat and habit. (B and C) Stems supported by numerous branched stilt roots. (D) Branches with pendulous aerial roots. (E) Mucronate leaf apex. (F) Leaf lamina with numerous, scattered tiny black cork warts on the abaxial surface. (G) Terminal interpetiolar stipule. (H) Terminal interpetiolar stipule with dense colleters inside at the base; colleters with a sticky white exudate. (I) Leaf scar with many tiny vascular bundle strands, arranged in a crescent moon. (J) Petiole with many tiny vascular bundle strands, arranged in a crescent moon. (K and L) Flowering branchlets showing inflorescences with flower buds and opened flowers. (M) Flower bud in longitudinal section. (N) Flowering and fruiting branchlets showing young inflorescences and mature fruits with persistent sepals. (O–R) Fruiting branchlets showing successive stages in development of hypocotyl. (S and T) Fruits showing successive stages in development of hypocotyl. (U) Fruits with seedling, hypocotyl disarticulated from cotyledonous cylindrical tube, revealing plumule. Photos: Chatchai Ngernsaengsaruay (A–L and N–U); Pichet Chanton (M).

Distribution. Africa [Tropical Africa: Egypt, Sudan, Eritrea, Djibouti, Somalia, Congo, Uganda, Kenya, Tanzania, Mozambique, Cape provinces, Natal, Aldabra, Comoros, Mozambique Channel, Madagascar, Seychelles], Mauritius (Island of France), Iran, Saudi Arabia, Yemen, Pakistan, India [Odisha, Andhra Pradesh, Tamil Nadu, Andaman and Nicobar Islands], Bangladesh [Chittagong, Sundarbans], Sri Lanka, Maldives, Myanmar [Arakan, Bago, Mergui Archipelago], China [Hainan], Taiwan, Japan [Ryukyu Islands (also called Nansei Islands), Okinawa Islands, Yaeyama Islands], Vietnam, Cambodia, Thailand, Peninsular Malaysia [Langkawi, Perak, Selangor, Pulau Indah (formerly known as Pulau Lumut), Terengganu (also called Trengganu),], Singapore, Indonesia [Sumatra, Java, Lesser Sunda Islands, Sulawesi, Moluccas, New Guinea], Borneo [Malaysia (Sarawak, Sabah), Brunei, Indonesia (Kalimantan)], Philippines [Luzon, Oriental Mindoro, Visayan Islands, Surigao del Norte, Sulu Archipelago, Palawan], Australia (Northern Territory, Queensland), Micronesia [Marianas Islands, Guam, Palau Islands, Caroline Islands, Marshall Islands], Melanesia [Papua New Guinea, Solomon Islands, Fiji, New Caledonia].

Distribution in Thailand. South-Western: Phetchaburi, Prachuap Khiri Khan; Central: Bangkok, Samut Prakan, Samut Songkhram, Samut Sakhon; South-Eastern: Chachoengsao, Chon Buri, Rayong, Chanthaburi, Trat; Peninsular: Chumphon, Ranong, Surat Thani, Phangnga, Phuket, Krabi, Nakhon Si Thammarat, Phatthalung, Trang, Satun, Songkhla, Pattani, Narathiwat.

Habitat and Ecology. Rhizophora mucronata is restricted to mangrove forests, gregarious on the muddy shorelines of estuaries, especially along riverbanks or creeks flooded by high tides, where it is often the dominant species.

Phenology. Flowering, fruiting and viviparous germination more than once, nearly throughout the year.

Conservation status. This species is very widespread from Africa, China to Tropical Asia, Australia to Micronesia and Melanesia. It is known from many localities and has a large EOO of 168,581,143.02 km2 and a relatively large AOO of 1,444 km2. Therefore, we consider the conservation assessment here as LC in agreement with Duke et al. (2010).

Etymology. The specific epithet of Rhizophora mucronata is a Latin word meaning ending abruptly in a hard short point (Stearn, 1992; Gledhill, 2002) and refers to a mucronate leaf apex formed by extending midrib.

Vernacular name. Kongkon (กงกอน) (Chumphon, Phetchaburi); Kongkang nok (กงกางนอก) (Phetchaburi); Kong keng (กงเกง) (Peninsular); Kongkang bai yai (โกงกางใบใหญ่) (Central); Phangka bai yai (พังกาใบใหญ่) (Peninsular); Asiatic mangrove, Red mangrove (English).

Uses. The wood is used for firewood and for making charcoal. Wood poles are used for piling and construction purposes. The timber is used for making furniture. The bark contains tannin, used for tanning leather and used for dyeing, toughening fishing nets, lines and ropes.

Lectotypifications. Griffith (1836: 8) established Rhizophora macrorrhiza based on the material collected from Mergui, he did not choose a holotype. However, we located the material W. Griffith s.n. from Mergui at G [G00441352]; therefore, this material is selected here as the lectotype, following Art. 9.3 and 9.12 of the ICN (Turland et al., 2018).

Blanco (1837: 398) erected Rhizophora longissima based on the specimen collected from Philippines but he did not mention the collector number and the name of the herbarium where the material was deposited. However, we could locate only one specimen E. D. Merrill Species Blancoanae 409 from Taytay, Palawan, Philippines housed in US [US00623705]. Blanco’s species was described in 1837, Merrill’s Species Blancoanae collected in 1913, so the latter was not present in 1837, is therefore not original material (Art. 9.4 of the ICN) (Turland et al., 2018) and cannot be lectotype (lectotypes can only be chosen from among the original material). Hence, this specimen is considered here as a neotype (Art. 9.13 of the ICN) (Turland et al., 2018).

Rhizophora latifolia was named by Miquel (1860: 324), who cited the specimen collected from Sumatra, “Tapanuli prope Djago Djago” [“T. (Teysmann)” = J. E. Teijsmann] but he did not mention the collector number and the name of the herbaria where the specimens were housed. We could trace two sheets of the specimen J. E. Teijsmann HB1064 (HB = Herbarium Buitenzorg, now Herbarium Bogoriense) from Tapanuli at L [L0009920, U0005822] and two sheets of the specimen J. E. Teijsmann HB1061 from Djago Djago at L [L0009921, U0005823] and following Art. 9.6 of the ICN (Turland et al., 2018), they constitute syntypes. Therefore, the L [U0005822] specimen is in the best condition and clearly shows the diagnostic characters for the species and is selected here as the lectotype, with an isolectotype at L [L0009920], following Art. 9.3 and 9.12 of the ICN (Turland et al., 2018).

Notes. The vernacular name “Kongkang bai yai” meaning big-leaved kongkang or big-leaved Rhizophora. The leaves of this species are usually bigger than Kongkang bai lek (Rhizophora apiculata).

Rhizophora mucronata is closely related to R. stylosa. It differs from the latter by free part of the ovary high conical, 1.5–3 mm long, in anthesis emerging far beyond the disk (vs free part of the ovary depressed-conical, hardly up to 1.5 mm long, in anthesis enclosed by the disk); style very short, up to 1.5 mm long (vs style filiform, 4–6 mm long); the rather larger leaves, 13–23.5 × 6–13 cm (vs the rather smaller leaves, 6.5–12.5 × 3.5–6.5 cm); and the petals with shorter hairs along the margins (vs the petals with longer hairs along the margins). The characters of R. stylosa were taken from Hou (1958, 1960). Furthermore, they also differ in ecology: R. mucronata is generally gregarious near and on the hanks of tidal creeks and on deep soft mud of estuaries while R. stylosa is exclusively found along sandy shores and on sand-covered coral terraces facing the open sea (Hou, 1958, 1960). R. stylosa distributed from China to Vietnam and Pacific (Hou, 1958; POWO, 2023).

Additional specimens examined. Thailand. South-Western: Phetchaburi [Hat Chao Samran subdistr., 26 May 2007, C. Ngernsaengsaruay own observation]; Prachuap Khiri Khan [Bang Saphan Yai distr., fl., 8 Nov 1943, T. Sindhiphong 80 (L [L2495789]); Khlong Wan, vivipary, 22 Oct 1964, C. Chermsirivathana 131 (BK); Sam Roi Yot (originally “Ratchaburi” on the label), fl. (L), vivipary (BKF), seedling (P), 7 Aug 1966, K. Larsen, T. Smitinand & E. Warncke 1223 (AAU, BKF [SN103705], L [L2495760], P [P05575595]); Khao Sam Roi Yot National Park, fl., 5 Dec 1979, T. Shimizu et al. T-26165 (BKF [SN049162]); Khlong Khao Daeng, Khao Sam Roi Yot National Park, Kui Buri distr., 28 May 2007, C. Ngernsaengsaruay own observation; Khlong Wan subdistr., 10 Jun 2007, C. Ngernsaengsaruay own observation; Khlong Ram Phueng, Bang Saphan distr., 18 Aug 2007, C. Ngernsaengsaruay own observation; Wat Khao Daeng, Kui Buri distr., fl., fr., 19 Sep 2017, P. Srisanga et al. 4083 (BKF [SN244628])]; Central: Bangkok [Bang Khun Thian, 23 Jun 2019, C. Ngernsaengsaruay own observation; Bang Khun Thian, reported by Consultants of Technology Co., Ltd. & Panya Consultants Co., Ltd. (2007)]; Samut Prakan [Ban Bang Pu, fl., 14 Jun 1968, H. M. van der Kevie 8A (L [L2495793]); Bang Kachao, Song Khanong subdistr., Phra Pradaeng distr., 30 Mar 2012, C. Ngernsaengsaruay own observation]; Samut Songkhram [Mueang Samut Songkhram distr., vivipary, 23 Aug 1975, J. F. Maxwell 75-911 (BK, L [L2495795]); Khlong Khon distr., 18 Jul 2016, C. Ngernsaengsaruay own observation]; Samut Sakhon [Ao Mahachai, 19 Jul 2016, C. Ngernsaengsaruay own observation]; South-Eastern: Chachoengsao [Bang Pakong distr., 18 Jan 2019, C. Ngernsaengsaruay own observation; Bang Pakong distr., reported by Lumsun, Kutintara & Emphandhu (2013)]; Chon Buri [reported by The Committee of Marine & Coastal Resources, Chonburi Province (2020)]; Rayong [Thung Kha Beach, Klaeng distr., reported by Chiwapreecha, Sootanan & Chantarasuwan (2021)]; Chanthaburi [Wan Chao, fl., Nov 1924, Unreadable s.n. (BK SN214354); Klung distr., vivipary, 18 May 1980, K. Iwatsuki T-27782 (BKF [SN049152]); Klung distr., fl., vivipary, 9 Dec 1983, N. Fukuoka & M. Ito T-34909 (BKF [SN099910], L [L4198575]); Klung distr., vivipary, 26 Sep 1984, N. Fukuoka T-36285 (BKF [SN049146], L [L4207254]); Khung Kraben, fl., 7 Feb 1991, K. Chayamarit & C. Phathanacharoen 210 (BKF [SN125744, SN125745]); the International Mangrove Botanical Garden Rama IX, Nong Bua subdistr., Mueang Chanthaburi distr., fl., fr., vivipary, 23 Sep 2023, C. Ngernsaengsaruay, P. Chanton & P. Aksornwachpalin IMBG004, IMBG005, IMBG006]; Trat [Laem Ngop distr., fl., 17 Feb 1923, U. Treesukon s.n. (BKF [SN049148]); Locality not specified, fl., 19 Jan 1958, T. Sørensen, K. Larsen & B. Hansen 534 (L [L2495786]); Ban Salak Khok, Ko Chang, 10 Jul 2022, H. Balslev et al. 10203 (AAU); Ko Kut, 6 Jul 2012, C. Ngernsaengsaruay own observation]; Peninsular: Chumphon [Unreadable, vivipary, 12 Jan 1927, A. F. G. Kerr 12882 (BK); Tako, Lang Suan distr., fl., 18 Jun 1928, P. Praisurin (Put) 1716 (BK); Khlong Tako, Lang Suan, fl., 14 Sep 1934, Smanwanakit 61 (L [L2495788]) Ban I Let, Hat Sai Ri subdistr., Mueang Chumphon distr., 14 May 2008, C. Ngernsaengsaruay own observation; Mu Ko Chumphon National Park, Hat Sai Ri subdistr., Mueang Chumphon distr., 14 May 2008, C. Ngernsaengsaruay own observation]; Ranong [Locality not specified, fl., 5 May 1968, C. F. van Beusekom & C. Phengklai 586 (E [E01058157], L [L2495763], P [P05575580]); Ban Pak Nam, fl., 9 Jul 1968, H. M. van der Kevie 22 (L [L2495792]); Locality not specified, sterile, Aug 1973, P. Nitrasirirak & C. Phengklai s.n. (BKF [SN049177], L [L2495764]); Kapoe distr., fl., 6 Feb 1982, Anon. s.n. (BKF [SN092772]); Kapoe distr., fl., vivipary, 25 Dec 1983, T. Santisuk 854 (BKF [SN174578]); Kapoe distr., fl., vivipary, 25 Dec 1983, N. Fukuoka & M. Ito T-35536 (BKF [SN049161], L [L4198571]); Locality not specified, fl., 10 Sep 1984, N. Fukuoka & W. Nanakhon T-36055 (BKF [SN097988]); Laem Son National Park, vivipary, 10 May 1990, R. Pooma 77 (BKF [SN171303]); Ngao subdistr., Mueang Ranong distr., vivipary, 30 Oct 1996, C. Dunbar 1 (L [L4192240]); Khlong Kamphuan, Andaman Coastal Research Station for Development, Kamphuan subdistr., Suk Samran distr., 10 Jul 2009, C. Ngernsaengsaruay own observation]; Surat Thani [Chaiya distr., fl., 11 Feb 1932, C. Bunchu 61 (BKF [SN049179, SN199623]); Don Sak distr., fl., vivipary, 1 Jan 1984, N. Fukuoka & M. Ito T-35701 (BKF [SN099914], L [L4210917]); Kadae subdistr., Kanchanadit distr., 18 May 2008, C. Ngernsaengsaruay own observation; Phum Riang subdistr., Chaiya distr., 19 May 2008, C. Ngernsaengsaruay own observation; Ko Samui, 17 May 2008, C. Ngernsaengsaruay own observation]; Phangnga [Thong Lang Village, Takua Thung distr. (originally “Phuket” on the label), fl., 29 Jan 1966, E. Hennipman 3751 (L [L2495798, L2495799], P [P05575599]); Ko Surin, 19 Apr 1976, C. Chermsirivathana & T. Smitinand 2129 (BK); Mu Ko Surin National Park, fl., 20 Feb 1995, T. Santisuk et al. s.n. (BKF [SN092395]); Ko Phe, Khura Buri distr., fl., 26 Apr 2005, C. Phengklai et al. 14964 (BKF [SN148564, SN148565]); Ko Kho Khao, Takua Pa distr., 19 Jun 2009, C. Ngernsaengsaruay own observation; Ko Panyi subdistr., Mueang Phangnga distr., 19 Jun 2009, C. Ngernsaengsaruay own observation]; Phuket [Locality not specified, fl., Apr 1915, Vanpruk 770 (BKF [SN049180])]; Krabi [Laem Nang, fl., 13 Nov 1966, B. Hansen & T. Smitinand 12346 (BKF [SN093206], E [E01058155]); Klong Prasong subdistr., Mueang Krabi distr., 21 Jun 2009, C. Ngernsaengsaruay own observation]; Nakhon Si Thammarat [Khlong Khanom Estuary, Khanom distr., 23 Jul 2010, C. Ngernsaengsaruay own observation; Pak Phanang Estuary, Pak Phanang distr., 24 Jul 2010, C. Ngernsaengsaruay own observation]; Phatthalung [Pak Phayun distr., 8 Mar 2022, C. Ngernsaengsaruay own observation]; Trang [Ban Yong Sata, vivipary, Sep 1924, Unreadable s.n. (BK SN214357); Kantang distr., fr., 20 Sep 1933, P. Praisurin 294 (BKF [SN049175]); Mai Fat subdistr., Sikao distr., fl., 27 Jan 2018, J. Sittichoke 6 (BKF [SN214531])]; Satun [Ra Wi, fl., 13 Jan 1928, A. F. G. Kerr (BK); Ban Hat Sai Khao, Tan Yong Po subdistr., Mueang Satun distr., 25 Oct 2010, C. Ngernsaengsaruay own observation; Ban Lom Puen, La-ngu distr., 26 Oct 2010, C. Ngernsaengsaruay own observation]; Songkhla [Tha Sa-an, Pha Wong subdistr., Mueang Songkhla distr., 23 Oct 2010, C. Ngernsaengsaruay own observation]; Pattani [Nong Chik distr., 4 Jun 2021, C. Ngernsaengsaruay own observation]; Narathiwat [Mueang Narathiwat distr., 3 Jun 2021, C. Ngernsaengsaruay own observation].

Measurements of the vegetative and reproductive parts of Rhizophora in Thailand are presented in Table 2.

Table 2 Measurements of the vegetative and reproductive parts of Rhizophora in Thailand.

Measurements of morphological characteristics	Sample sizes	Rhizophora apiculata	Rhizophora mucronata	
Ranges	Mean ± SD	Ranges	Mean ± SD	
Stipule length (cm)(forming a narrowly conical)	100	4.5–10.0	7.04 ± 0.95	4.5–12.5	8.16 ± 1.33	
Stipule diameter at the base (cm)	100	0.3–1.0	0.60 ± 0.17	0.5–1.7	0.93 ± 0.22	
Colleter length (mm)	50	0.3–1.0	0.65 ± 0.15	0.5–2.2	1.25 ± 0.45	
Colleter diameter at the base (mm)	50	0.1–0.2	0.14 ± 0.03	0.1–0.4	0.24 ± 0.06	
Colleter length/diameter ratio	50	2.1–7.7	3.00 ± 1.24	2.8–7.0	5.27 ± 1.25	
Leaf lamina length (cm)	200	12.0–20.5	15.84 ± 1.58	13.0–23.5	16.33 ± 1.78	
Leaf lamina width (cm)	200	5.0–8.5	6.16 ± 0.73	6.0–13.0	8.55 ± 1.22	
Leaf lamina length/width ratio	200	2.1–3.3	2.59 ± 0.25	1.6–2.3	1.92 ± 0.11	
Leaf lamina apex length (mm)	100	1.2–2.8	2.08 ± 0.31	3.2–7.0	5.11 ± 0.77	
Number of secondary veins per side	100	8–14	11.55 ± 1.42	9–14	11.20 ± 1.18	
Petiole length (cm)	200	1.3–4.0	2.48 ± 0.40	1.7–4.6	3.39 ± 0.52	
Petiole diameter at the middle (mm)	200	3.0–5.3	4.17 ± 0.44	3.0–6.0	4.18 ± 0.55	
Inflorescence length (cm)	100	1.6–3.0	2.30 ± 0.27	4.0–11.2	6.95 ± 1.78	
Peduncle length (cm)	100	0.4–1.4	0.73 ± 0.16	2.3–7.0	3.77 ± 1.01	
Peduncle diameter at the middle (mm)	100	3.5–8.5	5.57 ± 0.79	2.0–4.5	2.89 ± 0.48	
Number of flowers per mature inflorescence	50	2	2.00 ± 0.00	3–9	5.44 ± 1.49	
Number of flowers per young inflorescence	50	2	2.00 ± 0.00	5–16	8.88 ± 2.50	
Mature flower bud length (cm)	100	0.9–1.6	1.29 ± 0.17	1.1–1.7	1.39 ± 0.14	
Mature flower bud diameter at the widest (cm)	100	0.7–1.2	0.92 ± 0.11	0.6–1.0	0.79 ± 0.07	
Mature flower bud length/diameter ratio	100	1.2–1.7	1.41 ± 0.10	1.3–2.4	1.76 ± 0.19	
Open flower diameter (cm)	50	1.3–2.2	1.75 ± 0.23	1.4–2.0	1.68 ± 0.16	
Pedicel length (cm)	50	Sessile, with fleshy bracteoles at the base of the flower	0.6–1.5	0.88 ± 0.18	
Pedicel diameter (mm)	50			2.5–4.0	3.22 ± 0.40	
Sepal length (cm)	50	0.9–1.3	1.10 ± 0.11	0.9–1.2	1.01 ± 0.06	
Sepal width (cm)	50	0.4–0.7	0.56 ± 0.08	0.5–0.7	0.55 ± 0.04	
Sepal length/width ratio	50	1.5–2.8	2.00 ± 0.30	1.5–2.1	1.85 ± 0.13	
Petal length (cm)	50	0.5–1.2	0.85 ± 0.13	0.5–0.8	0.74 ± 0.05	
Petal width (cm)	50	0.1–0.2	0.14 ± 0.02	0.2–0.3	0.26 ± 0.03	
Petal length/width ratio	50	3.6–7.8	6.07 ± 1.01	2.2–3.7	2.84 ± 0.35	
Number of stamens per flowers	50	(8–)12(–14)	11.50 ± 1.66	8.0	8.00 ± 0.00	
Anther length (mm)	50	5.8–9.5	7.57 ± 0.83	5.5–7.8	6.67 ± 0.44	
Anther width (mm)	50	1.3–3.0	2.20 ± 0.45	1.4–2.2	1.81 ± 0.16	
Free part of the ovary length (mm)	50	2.0–3.4	2.80 ± 0.37	1.5–3.0	2.12 ± 0.38	
Free part of the ovary diameter (mm)	50	1.6–3.7	2.61 ± 0.41	1.8–3.6	2.50 ± 0.44	
Number of radiate ridges on free part of the ovary	50	7.0–10.0	8.18 ± 0.83	7.0–9.0	7.98 ± 0.55	
Infructescence stalk length (cm)	80	0.8–2.3	0.89 ± 0.24	2.4–6.0	3.99 ± 0.81	
Infructescence stalk diameter (mm)	80	3.5–9.0	5.10 ± 0.73	2.5–4.5	3.29 ± 0.41	
Fruit length (cm) (before seed germination)	50	1.5–2.6	2.05 ± 0.26	3.7–5.0	4.50 ± 0.31	
Fruit diameter at the base (cm)	50	1.0–1.4	1.22 ± 0.10	2.1–2.8	2.39 ± 0.18	
Fruit length/diameter ratio	50	1.4–2.0	1.68 ± 0.15	1.5–2.3	1.90 ± 0.21	
Fruit length (cm) (hypocotyls nearly come off)	100	2.2–3.3	2.49 ± 0.23	4.0–7.0	6.04 ± 0.47	
Fruit diameter at the basal part (cm)	100	1.2–1.8	1.48 ± 0.13	2.4–3.5	3.05 ± 0.19	
Fruit diameter at the apical part (cm)	100	0.8–1.2	1.04 ± 0.07	1.1–1.9	1.49 ± 0.19	
Persistent sepal length (cm)	100	1.1–1.9	1.37 ± 0.13	0.9–1.4	1.17 ± 0.12	
Persistent sepal width (cm)	100	0.6–0.8	0.71 ± 0.04	0.5–0.9	0.74 ± 0.08	
Fruit stalk length (cm)	100	Sessile, with fleshy bracteoles at the base of the fruit	0.5–1.3	0.79 ± 0.17	
Fruit stalk diameter (mm)	100			3.5–6.0	4.45 ± 0.50	
Cotyledonous tube diameter (cm)	100	0.5–0.8	0.64 ± 0.08	0.7–1.0	0.82 ± 0.09	
Hypocotyl length (cm)	100	25.0–45.0	32.40 ± 4.87	25.0–70.0	46.16 ± 7.04	
Hypocotyl diameter at the middle part (cm)	100	0.6–1.1	0.86 ± 0.09	0.9–1.6	1.27 ± 0.16	
Hypocotyl diameter at the basal part (cm)	100	0.8–1.3	1.13 ± 0.12	0.7–1.7	1.13 ± 0.19	

Anatomical study

Leaf anatomy of Thai Rhizophora species

The leaf anatomical features of these two Rhizophora species consist of five major tissue layers, i.e., adaxial (upper) epidermis, hypodermis, palisade mesophyll (also called palisade parenchyma), spongy mesophyll (also called spongy parenchyma), and abaxial (lower) epidermis. The leaves of R. mucronata (456.80–537.41 µm) are thicker than R. apiculata (375.46–456.11). The leaf epidermis of Rhizophora is covered by the thick cuticular wax (cuticle) layer. The adaxial surface has a slight thicker cuticle than the abaxial surface (Figs. 10A, 10C, 11A and 11C). As the results of Vinoth, Kumaravel & Ranganathan (2019) and Putri & Bashri (2023) revealed that the cuticles on Rhizophora leaves which play a role in reducing excessive evaporation or transpiration of water by the leaves and the thickness of the cuticles which play a significant role in retaining water to support plant development in saline conditions.

Figure 10 Leaf anatomy of Rhizophora apiculata. (A–C) Transverse section. (D) Abaxial epidermis (under LM). (E) Abaxial epidermis (under SEM) (Abe, abaxial epidermis; Ade, adaxial epidermis; Cr, druse crystal; Ct, cuticle; Cw, cork wart; EpC, epidermal cell; Hp, hypodermis; P, phloem; Pa, parenchyma; Pl, palisade mesophyll; Sp, spongy mesophyll; St, stoma; SubC, subsidiary cell; Vb, vascular bundle; X, xylem). Photos: Pichet Chanton & Chatchai Ngernsaengsaruay.

Figure 11 Leaf anatomy of Rhizophora mucronata. (A–C) Transverse section. (D) Abaxial epidermis (under LM). (E) Abaxial epidermis (under SEM) (Abe, abaxial epidermis; Ade, adaxial epidermis; Cr, druse crystal; Ct, cuticle; Cw, cork wart; EpC, epidermal cell; Hp, hypodermis; P, phloem; Pa, parenchyma; Pl, palisade mesophyll; Sp, spongy mesophyll; St, stoma; SubC, subsidiary cell; Vb, vascular bundle; X, xylem). Photos: Pichet Chanton & Chatchai Ngernsaengsaruay.

The epidermal cells of the two Rhizophora species are polygonal, sometimes trapezium in shape with straight anticlinal walls and arranged in a single layer on both adaxial and abaxial leaf surfaces. The adaxial surface has a slight thicker epidermal layer than the abaxial surface (Figs. 10C, 10D, 11C and 11D).

The stomata of Rhizophora are confined only to the abaxial leaf surface (hypostomatic leaves) and are sunken cyclocytic, which are surrounded by 6–8 subsidiary cells arranged in a ring around the stoma (Figs. 10C–10E and 11C–11E). The stomatal size of these two Rhizophora species is 17.42–30.38 × 9.20–16.28 µm and the number of stomata is 155–358 stomata/mm2. R. apiculata [206–358 stomata/mm2 (275.21 ± 41.67)] has a higher stomatal density than R. mucronata [155–264 stomata/mm2 (203.67 ± 31.34)]. Based on previous study, the stomatal density is classified as three groups: low density (<300 mm2), medium density (300–500/mm2), and high density (>500/mm2). Therefore, the stomatal density of these two Rhizophora species is classified as low to medium. This is supported by the statement of Juairiah (2014), the low density of stomata is one of the anatomical adaptations of Rhizophora and can prevent excessive water transpiration so that Rhizophora leaves have enough water content considering that this plant lives in an environment with high levels of salt and sunlight. As mentioned by Tomlinson (1986, 2016), the sunken stomata are an adaptation to the mangrove environment as the xerophytic plants.

The number of cork warts of Thai Rhizophora species is 14–115 cork warts/cm2. R. apiculata [29–115 cork warts/cm2 (66.79 ± 22.79, n = 300)] has a higher cork warts density than R. mucronata [14–50 cork warts/cm2 (29.42 ± 8.62, n = 300)]. R. apiculata and R. mucronata have a higher average cork wart density at the apical part than the middle and the basal parts of leaves (Figs. 10E and 11E).

The cork warts are one of the characteristics of the anatomical adaptation of leaves in plants that live in high-salinity habitats and function as a medium for excessive salt excretion in leaves (Baijnath & Charles, 1980; Putri & Bashri, 2023). Moreover, the proposed path of internal airflow in Rhizophora mangle is (1) air enters leaves via cork warts, (2) air moves into the leaf aerenchyma, (3) air moves into petiole aerenchyma, (4) air moves into the centrally-located aerenchyma of stem. (5) The air descends through the aerenchyma of stems and into the inner aerenchyma of stilt (air) roots, (6) and into the root terminals of mud roots. (7) From there, air ascends the outer aerenchyma of the same roots, and (8) is released to the environment via lenticels (Evans, Okawa & Searcy, 2005).

The Rhizophora are true mangrove plants which do not have salt glands on their leaves. This is supported by the statement of Putri & Bashri (2023); they belong to the non-secretor true mangrove species.

The two Rhizophora species have bifacial leaves (also called dorsiventral leaves). The mesophyll composed of hypodermis, palisade parenchyma, and spongy parenchyma (Figs. 10A, 10C, 11A and 11C).

Hypodermis of Thai Rhizophora species are usually large cells, arranged in multiple layers below epidermis and often filled with tannins so that presenting as dark stained. The adaxial surface has a higher average hypodermal layer thickness than the abaxial surface. The hypodermal layer thickness on both adaxial and abaxial surfaces of R. mucronata is thicker than R. apiculata. The shape of hypodermal cells on the adaxial surface can be broadly elliptic, elliptic, narrowly elliptic, oblong, subcircular or circular and the abaxial surface can be broadly elliptic, subcircular or circular. The adaxial surface has a larger average hypodermal cell size than the abaxial surface. The hypodermal cell size on both adaxial and abaxial surfaces of R. mucronata is larger than R. apiculata (Figs. 10A, 10C, 11A and 11C).

Druse crystals are found in hypodermal cells on both adaxial and abaxial sides. The diameter of the druse crystals is 8.92–26.54 µm (Figs. 10C and 11C).

According to previous studies, these hypodermis cells are generally found in halophytic plants, which have watery habitats with high salt content. The hypodermis has a role as a water storage tissue (Tomlinson, 1986, 2016) and also used by Rhizophora as a salt storage tissue, which is absorbed by the plants. Salt absorbed by the roots will be collected in these hypodermal cells, which will later be released by the Rhizophora plant when the leaves come off (Vinoth, Kumaravel & Ranganathan, 2019; Putri & Bashri, 2023). The size of these hypodermal cells influences the effectiveness of water storage in the leaves of each species (Putri & Bashri, 2023).

Based on this study, the thick cuticles, sunken stomata and large hypodermal cells are adaptive features of leaves of Rhizophora and could be concluded as a range of xeromorphic features as mentioned by Sheue (2003).

Palisade parenchyma consist of tightly packed layer of elongated cells beneath the hypodermis. Spongy parenchyma composed of loosely packed layer of irregularly shaped cells beneath the palisade parenchyma, underlying the lower epidermis, with prominent intercellular space, and represent as netted work. The number of palisade cell layer of these two Rhizophora species is 1–2-layered and the number of spongy cell layer is 5–8-layered (Figs. 10A, 10C, 11A and 11C).

Observations on the midrib vascular bundles arrangement has shown that Rhizophora have complex vascular bundles and are divided into three parts: abaxial part, medullary (middle) part, and adaxial part. The vascular bundles in the midrib consist of phloem outside and xylem inside, and these vascular bundles are incompletely enclosed by a layer of sclerenchyma cells (Figs. 10A, 10B, 11A and 11B). The presence of sclerenchyma cells in the midrib vascular bundles is to provide support and protection for the leaf structure.

A comparison of leaf anatomical characters of Rhizophora in Thailand with previous studies is shown in Table 3.

Table 3 A comparison of leaf anatomical characters of Rhizophora in Thailand with previous studies of other countries.

The leaf anatomical characters of Rhizophora apiculata and R. mucronata in other countries were taken from previous studies, [1] Sheue (2003), [2] Vinoth, Kumaravel & Ranganathan (2019), and [3] Putri & Bashri (2023).

Characters	From the author’s observations	Previous studies	
Rhizophora apiculata	Rhizophora mucronata	Rhizophora apiculata	Rhizophora mucronata	
Leaf thickness (µm)	375.46–456.11 (n = 20)	456.80–537.41 (n = 20)	464.6 ± 22.2 (India) [1]	700.0 ± 21.0 (India) [1]	
(414.26 ± 23.22)	(494.45 ± 23.96)	504.3 ± 83.0 (Australia) [1]	475.3 ± 160.0 (Singapore) [1]	
Leaf structure types	Bifacial leaves	Bifacial leaves	Dorsiventral leaves [1] (=bifacial leaves)	Dorsiventral leaves [1]	
Cuticular wax thickness on the adaxial leaf surface (µm)	4.88–12.84 (n = 20)	6.38–10.63 (n = 20)	6.37 ± 1.13 [1]	4.60 ± 1.10 [1]	
(7.72 ± 2.28)	(8.83 ± 1.20)	Present [3]	Present [3]	
Cuticular wax thickness on the abaxial leaf surface (µm)	4.06–10.56 (n = 20)	6.13–8.47 (n = 20)	3.76 ± 1.10 [1]	4.42 ± 1.16 [1]	
	(7.41 ± 2.05)	(7.68 ± 0.71)	Absent [3]	Absent [3]	
Number of epidermal cell layer on the adaxial leaf surface	1-layered	1-layered	1-layered [1]	1-layered [1]	
Number of epidermal cell layer on the abaxial leaf surface	1-layered	1-layered	1-layered [1]	1-layered [1]	
Epidermal layer thickness on the adaxial leaf surface (µm)	10.42–18.88 (n = 20)	13.59–20.34 (n = 20)	–	–	
(14.33 ± 2.72)	(17.37 ± 1.87)	
Epidermal layer thickness on the abaxial leaf surface (µm)	9.58–16.50 (n = 20)	10.82–19.44 (n = 20)	–	–	
(13.21 ± 2.32)	(14.89 ± 2.67)	
Epidermal cell shapes on the adaxial leaf surface	Polygonal, sometimes trapezium	Polygonal, sometimes trapezium	Polygonal [2], [3]	Polygonal [2], [3]	
Epidermal cell shapes on the abaxial leaf surface	Polygonal, sometimes trapezium	Polygonal, sometimes trapezium	Polygonal [3]	Polygonal [3]	
Stomatal types	Sunken cyclocytic	sunken cyclocytic	Cyclocytic [1], [2]	Cyclocytic [1], [2]	
Stomatal density per mm2	206–358 (n = 100)	155–264 (n = 100)	12–15 (number of stomata in one field of view, 400x magnification) [3]	9–12 (number of stomata in one field of view, 400x magnification) [3]	
(275.21 ± 41.67)	(203.67 ± 31.34)	
66.24 (mean), low density [3]	50.96 (mean), low density [3]	
Stomatal length (µm)	17.46–30.38 (n = 100)	17.42–28.99 (n = 100)	42.35 ± 1.67 [1]	27.87 ± 2.27 [1]	
(24.09 ± 3.84)	(22.68 ± 3.20)	41–50 [3]	51–56 [3]	
Stomatal width (µm)	9.20–16.28 (n = 100)	10.10–15.88 (n = 100)	42.90 ± 3.03 [1]	42.53 ± 1.80 [1]	
(13.01 ± 1.91)	(12.98 ± 1.62)	24–34 [3]	26–37 [3]	
Stomatal length/width ratio	1.08–3.11 (n = 100)	1.14–2.61 (n = 100)	1.52 [1]	1.00 [1]	
(1.90 ± 0.47)	(1.78 ± 0.34)	
Number of subsidiary cells	6–8 (n = 50)	6–8 (n = 50)	5–7(–8) [1]	(6–)7(–8) [1]	
(7.54 ± 0.76)	(6.62 ± 0.81)	
Characters	From the author’s observations	Previous studies	
Rhizophora apiculata	Rhizophora mucronata	Rhizophora apiculata	Rhizophora mucronata	
Cork wart density per cm2	29–115 (n = 300)	14–50 (n = 300)	–	–	
(66.79 ± 22.79)	(29.42 ± 8.62)	
Cork wart density at the apical part per cm2	70–115 (n = 100)	28–50 (n = 100)	–	–	
(92.60 ± 11.20)	(38.41 ± 5.47)	
Cork wart density at the middle part per cm2	52–80 (n = 100)	21–37 (n = 100)	–	–	
(66.74 ± 7.65)	(29.79 ± 3.39)	
Cork wart density at the basal part per cm2	29–52 (n = 100)	14–25 (n = 100)	–	–	
(41.04 ± 6.50)	(20.06 ± 2.98)	
Cork wart diameter (µm)	35.69–75.48 (n = 100)	39.86–82.36 (n = 100)	105.60 ± 13.25 (Singapore) [1]	149.05 ± 26.00 (Singapore) [1]	
(54.99 ± 10.64)	(61.76 ± 12.83)	
108.11 ± 31.20 (Australia) [1]	163.90 ± 29.60 (India) [1]	
Present [3]	Absent [3]	
Hypodermal layer thickness on the adaxial leaf surface (µm)	89.50–146.65 (n = 20)	130.11–200.67 (n = 20)	–	–	
(109.97 ± 13.49)	(155.38 ± 20.00)	
Hypodermal layer thickness on the abaxial leaf surface (µm)	19.18–29.86 (n = 20)	33.80–51.56 (n = 20)	–	–	
(24.00 ± 3.37)	(43.27 ± 5.84)	
Number of hypodermal cell layer on the adaxial leaf surface	3–6-layered (n = 20)	2–3-layered (n = 20)	2-layered (small-sized) [1] +	2(–3)-layered (small-sized) [1] +	
(4.50 ± 1.00)	(2.50 ± 0.51)	2-layered (large-sized) [1]	2(–3)-layered (large-sized) [1]	
4-layered [3]	6-layered [3]	
Number of hypodermal cell layer on the abaxial leaf surface	1–2-layered (n = 20)	2–3-layered (n = 20)	3–4-layered [1]	3-layered [1]	
(1.65 ± 0.49)	(2.30 ± 0.47)	Absent [3]	Absent [3]	
Hypodermal cell shapes on the adaxial leaf surface (µm)	Broadly elliptic, oblong, subcircular or circular	Broadly elliptic, elliptic, narrowly elliptic, subcircular or circular	Rounded hexagon [3]	Rounded hexagon [3]	
Hypodermal cell length on the adaxial leaf surface (µm)	25.13–63.88 (n = 20)	45.99–125.80 (n = 20)	32.26 ± 3.59 (upper group) [1]	45–60 [3]	
(41.35 ± 10.98)	(80.87 ± 22.42)	56.47 ± 8.33 (lower group) [1]	
54–69 [3]	
Hypodermal cell width on the adaxial leaf surface (µm)	23.30–50.80 (n = 20)	36.31–86.40 (n = 20)	16.13 ± 3.31 (upper group) [1]	36–51 [3]	
(36.10 ± 7.77)	(58.02 ± 11.99)	46.93 ± 10.40 (lower group) [1]	
38–57 [3]	
Hypodermal cell shapes on the abaxial leaf surface (µm)	Broadly elliptic, subcircular or circular	broadly elliptic, subcircular or circular	–	–	
Hypodermal cell length on the abaxial leaf surface (µm)	14.46–32.65 (n = 20)	18.61–39.72 (n = 20)	–	–	
(20.31 ± 5.05)	(26.82 ± 6.66)	
Hypodermal cell width on the abaxial leaf surface (µm)	13.57–29.07 (n = 20)	15.30–37.84 (n = 20)	–	–	
(20.08 ± 3.58)	(25.66 ± 7.47)	
Crystal location, type and diameter (µm)	In hypodermal cells, druse,	In hypodermal cells druse,	–	–	
8.92–26.54 (n = 20)	11.96–25.14 (n = 20)	
(16.76 ± 4.92)	(18.47 ± 4.50)	
Number of palisade cell layer	1–2-layered (n = 20)	1–2-layered (n = 20)	1–4-layered [1]	1–4-layered [1]	
(1.70 ± 0.47)	(1.45 ± 0.51)	
Number of spongy cell layer	5–8-layered (n = 20)	6–8-layered (n = 20)	4-layered (India) [1]	11-layered (India) [1]	
(6.55 ± 1.05)	(7.05 ± 0.76)	6–8-layered (Australia) [1]	8–9-layered (Singapore) [1]	

Palynological study

Pollen morphology of Thai Rhizophora species

The pollen grains of two species of Thai Rhizophora are monads, isopolar, radially symmetrical, tricolporate, and prolate spheroidal or oblate spheroidal in shape (P/E ratio = 0.96–1.12). The outline shape in polar view (amb) is subcircular or rounded-triangular. The size of the pollen grains is small, the polar axis diameter varies from 11.16–18.49 µm, and the equatorial axis diameter varies from 10.35–18.27 µm. The colpus length of the pollen grains ranges from 8.39–16.87 µm and the colpus width ranges from 1.30–4.01 µm. The exine thickness is 0.37–1.34 µm. and the exine sculpturing is reticulate (Fig. 12).

Figure 12 LM and SEM micrographs of pollen grains of Rhizophora in Thailand. (A and B) R. apiculata under LM. (A) Tricolporate aperture and rounded-triangular shape in polar view. (B) Colporate aperture in equatorial view. (C–E) R. apiculata under SEM. (C) Tricolporate aperture in polar view. (D) Colporate aperture in equatorial view. (E) Reticulate exine sculpturing. (F and G) R. mucronata under LM. (F) Tricolporate aperture and subcircular shape in polar view. (G) Colporate aperture in equatorial view. (H–J) R. mucronata under SEM. (H) Tricolporate aperture in polar view. (I) Colporate aperture in equatorial view. (J) Reticulate exine sculpturing. Photos: Pichet Chanton (under LM); Chatchai Ngernsaengsaruay & Pichet Chanton (under SEM).

A comparison of pollen morphology of Rhizophora in Thailand with previous studies is presented in Table 4.

Table 4 A comparison of pollen morphology of Rhizophora in Thailand with previous studies of other countries.

The pollen morphological characters of Rhizophora apiculata and R. mucronata in other countries were taken from previous studies, [1] Mao et al. (2012), [2] Mohd-Arrabe’ & Noraini (2013), and [3] Dalimunthe et al. (2023).

Characters	From the author’s observations	Previous studies	
Rhizophora apiculata	Rhizophora mucronata	Rhizophora apiculata	Rhizophora mucronata	
Polar axis [P] length (μm)	11.16–13.50 (12.36 ± 0.78) (n = 30)	12.37–18.49 (15.77 ± 1.58) (n = 30)	20.00–28.00 (mean 24.40) [1]	18.7–26.8 (mean 23.80) [1]	
17.38–21.05 (mean 19.16) [2]	19.16–25.95 (mean 22.83) [2]	
15.76–16.00 (15.88 ± 0.20) [3]	
Equatorial axis [E] length (μm)	10.35–12.92 (11.85 ± 0.68) (n = 30)	11.35–18.27 (15.69 ± 1.73) (n = 30)	19.60–27.80 (mean 22.60) [1]	16.6–24.4 (mean 20.40) [1]	
13.19–20.87 (mean 18.14) [2]	14.71–23.73 (mean 20.96) [2]	
24.68–25.08 (24.88 ± 0.20) [3]	
P/E ratio	0.98–1.12 (1.04 ± 0.04)	0.96–1.10 (1.01 ± 0.04)	1.06 [2]	1.09 [2]	
Pollen size classes	Small	Small	Small to medium [1]	Small to medium [1], [2]	
Small [2], [3]	
Pollen shapes	Prolate spheroidal or oblate spheroidal	Prolate spheroidal or oblate spheroidal	Prolate to spheroidal [1]	Prolate spheroidal [1], [2]	
Prolate spheroidal [2]	
Prolate [3]	
Pollen aperture	Tricolporate	Tricolporate	Mostly tricolporate, sometimes	Tricolporate [1], [2]	
Tetracolporate [1]	
Tricolporate [2], [3]	
Colpus length (μm)	8.39–10.86 (9.46 ± 0.65)	9.40–16.87 (12.87 ± 2.06)	ca. 12.60 [1]	ca. 15.80 [1]	
Colpus width (μm)	1.30–4.01 (2.43 ± 0.60)	2.17–3.82 (3.04 ± 0.48)	–	–	
Exine thickness (μm)	0.37–1.26 (0.70 ± 0.20)	0.55–1.34 (0.88 ± 0.24)	ca. 2.20 [1]	ca. 1.40 [1]	
0.73–1.48 (mean 1.02) [2]	0.60–1.95 (mean 1.03) [2]	
Exine sculpturing	Reticulate	Reticulate	Perforate to (irregularly)	Perforate [1]	
Perforate–reticulate [1]	Perforate to reticulate [2]	
Perforate to reticulate [2]	
Reticulate [3]	

Discussion

According to Hou (1958, 1970), Henslow (1878), Qin & Boufford (2007), and Schwarzbach (2014), the sepals of Rhizophora are united at the base forming a calyx tube and above with four lobes. From the author’s observations, we found four sepals are not united (free) but basally adnate to the rim of the hypanthium. Therefore, we suggest using four sepals but not 4-lobed calyx in agreement with McCusker (1984).

As stated by Sheue (2003), in the eastern mangrove (the areas between the Eastern Africa to the Pacific West Coast), the stipules of Rhizophora are rounded in outline in transverse section, and the colleters are acuminate rod in shape. However, from our examinations, we found the stipules can be depressed orbicular at the basal part, suborbicular to orbicular at the apical part in outline in transverse section, and the colleters are narrowly conical in shape.

In addition to keys to the species of Rhizophora as mentioned by Hou (1958) in Malesia, the region including Malaysia, Singapore, Indonesia, Brunei Darussalam, the Philippines, Timor-Leste, and Papua New Guinea and Hou (1970) in Thailand, Rhizophora apiculata differs from R. mucronata in its leaf lamina shape and width, a terminal stiff point length of the leaf laminae; inflorescence type; bracteole color; the presence or absence of a pedicel in flower; mature flower bud shape; sepal shape, color, and texture; petal shape and folding or flattened; stamen number; fruit size; hypocotyl color and size; and cotyledonous cylindrical tubes color and diameter.

According to previous studies of Rhizophora apiculata, the shape and size of leaf laminae are elliptic-oblong to sublanceolate and 7–18 × 3–8 cm (Hou, 1958, 1970; Qin & Boufford, 2007 in China) and elliptic or broadly lanceolate and 8–19 × 4–9 cm (McCusker, 1984 in Australia). The length of petioles is 1.5–3 cm (Hou, 1958, 1970; Qin & Boufford, 2007) and 1.5–4 cm (McCusker, 1984). The shape and length of stipules are lanceolate and 4–8 cm (Hou, 1958, 1970; Qin & Boufford, 2007). Furthermore, from our examinations, we found the laminae of this species are elliptic, narrowly elliptic or lanceolate-ovate and 12–20.5 × 5–8.5 cm, which are often smaller than those of R. mucronata; the length of petioles is 1.3–4 cm; and the shape and length of stipules are linear-lanceolate and 4.5–10 cm.

According to Hou (1958), the shape and size of mature flower buds of Rhizophora apiculata are elliptic and 1.4 cm long; however, from the author’s observations, we found the mature flower buds can be broadly ellipsoid or ellipsoid, widest at the middle and 0.9–1.6 × 0.7–1.2 cm.

According to previous studies of Rhizophora apiculata, the shape and size of mature fruits are obpyriform and 2–2.5 cm long (Hou, 1958), obpyriform and 2–3.5 cm long (Backer & Bakhuizen van den Brink, 1963 in Java, Indonesia), c. 2.5 × 1.5–cm (Qin & Boufford, 2007), and pyriform and 2–2.5 cm long (McCusker, 1984); however, from our examinations, we found the mature fruits of this species can be ovoid, sometimes broadly ovoid, 1.5–2.6 cm long, 1–1.4 cm in diam. at the base (before seed germination) and obpyriform, 2.2–3.3 cm long, 1.2–1.8 cm in diam. at the base (when the hypocotyls nearly come off), which are usually smaller than those of R. mucronata.

According to previous studies of Rhizophora apiculata, the shape and size of hypocotyls (before falling off) are cylindrical-clavate and 38 × 1.2 cm (Hou, 1958, 1970; Qin & Boufford, 2007), clavate and 25–30 cm long (Backer & Bakhuizen van den Brink, 1963), and clavate and 20–40 cm long (McCusker, 1984). Furthermore, from our study, we found the hypocotyls of this species are cylindrical-clavate and 25–45 cm long, middle part 0.6–1.1 cm in diam. and basal part 0.8–1.3 cm in diam, which are usually smaller than those of R. mucronata.

In addition to Hou (1970), the distribution of Rhizophora apiculata in Phetchaburi, Bangkok, Samut Prakan, Samut Songkhram, Samut Sakhon, Chachoengsao, Rayong, Ranong, Phangnga, Phuket, Krabi, Nakhon Si Thammarat, Phatthalung, Songkhla, Pattani, and Narathiwat provinces were newly recorded in this study.

According to previous studies of Rhizophora mucronata, the shape and size of leaf laminae are broadly elliptic to oblong and (8.5–)11–18(–23) × 5–10.5(–13) cm (Hou, 1958, 1970; Qin & Boufford, 2007) and broadly elliptic to obovate and 11–20 × 5–11 cm (McCusker, 1984). The length of petioles is 2.5–5.5 cm (Hou, 1958, 1970), 2–4 cm (McCusker, 1984), and 2.5–4 cm (Qin & Boufford, 2007). The shape and length of stipules are lanceolate and 5.5–8.5 cm (Hou, 1958, 1970; Qin & Boufford, 2007). However, from the author’s observations, we found the laminae of this species are ovate or lanceolate-ovate and 13–23.5 × 6–13 cm; the length of petioles is 1.7–4.6 cm; and the shape and length of stipules are linear-lanceolate 4.5–12.3 cm.

According to previous studies, the number of flowers per inflorescence of Rhizophora mucronata has a range of 2–5(–12) (Hou, 1958, 1970), 2–4(–8) (Qin & Boufford, 2007), and 2–4, occasionally one or up to 12 (McCusker, 1984). From our study, we found that the number of flowers per inflorescence of this species has a range of 3–9 (5–16 flowers in young inflorescences, some flowers falling off before mature).

According to Hou (1958), the size of mature flower buds of Rhizophora mucronata is widest near the base; however, from our examinations, we found the shape and size of mature flower buds can be ovoid, sometimes conical-ovoid and 1.1–1.7 cm long, 0.6–1 cm in diam.

According to previous studies of Rhizophora mucronata, the shape and size of mature fruits are elongate ovoid and 5–7 × 2.5–3.5 cm (Hou, 1958; Backer & Bakhuizen van den Brink, 1963; Qin & Boufford, 2007) and ovoid or pyriform and 4–6 cm long (McCusker, 1984); however, from our examinations, we found the mature fruits of this species can be ovoid or conical-ovoid, 3.7–5 cm long, 2.1–2.8 cm in diam. at the base (before seed germination); and obpyriform, 4–7 cm long, basally 2.4–3.5 cm in diam. and apically 1.1–1.9 cm in diam. (when the hypocotyls nearly come off).

According to previous studies of Rhizophora mucronata, the shape and size of hypocotyls (before falling off) are cylindrical and 36–64 × 1.8 cm (Hou, 1958, 1970), cylindrical and 30–60 cm long (Qin & Boufford, 2007), clavate and 40–60 cm long (Backer & Bakhuizen van den Brink, 1963), and 40–80 cm long (McCusker, 1984). Furthermore, from our study, we found the hypocotyls of this species are cylindrical-clavate and 25–70 cm long, middle part 0.9–1.6 cm in diam. and basal part 0.7–1.7 cm in diam.

In addition to Hou (1970), the distribution of Rhizophora mucronata in Phetchaburi, Bangkok, Samut Prakan, Samut Songkhram, Samut Sakhon, Chachoengsao, Chon Buri, Rayong, Trat, Ranong, Phangnga, Phuket, Krabi, Nakhon Si Thammarat, Phatthalung, Satun, Songkhla, Pattani, Narathiwat provinces were newly recorded in this study.

The tiny black dot-like structures, cork warts, are usually found on the lower leaf surface of Rhizophora. According to Hou (1958) and Tomlinson (1986, 2016) regarded the cork warts as a diagnostic feature for species of Rhizophora, but in North-Eastern Australia Duke & Bunt (1979) did not agree with their opinions because the cork warts could not be found in the leaves of R. apiculata in Australia. Furthermore, Sheue (2003) mentioned the cork warts actually could not be observed by naked eyes from the leaves of R. apiculata in the field in Australia as stated by Duke & Bunt (1979). However, the cork warts do exist while examining those leaves with light microscope after the leaves were treated with clearing method (Sheue, 2003). But, from our examinations, the cork warts can be observed by naked eyes from the leaves of R. apiculata and R. mucronata.

From the author’s observations, the cork wart diameter of R. apiculata is 35.69–75.48 (54.99 ± 10.64; n = 100) and R. mucronata is 39.86–82.36 (61.76 ± 12.83; n = 100). Nevertheless, as stated by Sheue (2003), R. apiculata (105.60 ± 13.25 µm in diam. from Singapore and 108.11 ± 31.20 µm in diam. from Australia) is the species with the smallest size of cork warts and R. styolosa (196.00 ± 20.86 µm in diam. from Singapore and 252.00 ± 41.11 µm in diam. from Taiwan) has the largest size of cork warts than those of R. mucronata (149.05 ± 26.00 µm in diam. from Singapore and 163.90 ± 29.60 µm in diam. from India).

In South Africa, Baijnath & Charles (1980) indicated the cork warts also present on the adaxial leaf surface of Rhizophora, but from our observations, we found Rhizophora have cork warts distributed only on the abaxial leaf surface in agreement with Sheue (2003).

According to Putri & Bashri (2023) in Banyuurip Mangrove Center, Indonesia, the stomata of Rhizophora apiculata and R. mucronata are paracytic, but from our examinations, we found the stomata of these two species are cyclocytic, which are surrounded by 6–8 subsidiary cells arranged in a ring around the stoma in agreement with Sheue (2003) and Vinoth, Kumaravel & Ranganathan (2019) in the east coast of Tamil Nadu, India.

From our observations, the average of stomatal density of Rhizophora apiculata (275.21 ± 41.67/mm2) is higher than those of R. mucronata (203.67 ± 31.34/mm2). Nevertheless, as stated by Putri & Bashri (2023), R. apiculata has an average stomatal density of 66.24/mm2 and R. mucronata has an average stomatal density of 50.96/mm2.

As stated by Putri & Bashri (2023), the stomatal size of Rhizophora apiculata (41–50 × 24–34 µm) is smaller than those of R. mucronata (51–56 × 26–37 µm). However, from our observations, we found the stomata of R. apiculata (17.46–30.38 × 9.20–16.28 µm) are the same size as R. mucronata (17.42–28.99 ×10.10–15.88 µm) and both species are smaller as a result of Putri & Bashri (2023).

As the results of Surya & Hari (2017) in Kerala, India, revealed that the size (P × E) of the pollen grains of Rhizophora apiculata and R. mucronata are 3.39–3.67 × 2.84–2.99 μm (P/E = 3.15–4.14) and 3.46–4.04 × 3.20–3.70 μm (P/E = 2.81–3.09), respectively. But, from our observations, the size of the pollen grains of these two Thai Rhizophora species are larger: R. apiculata with 11.16–13.50 (12.36 ± 0.78) × 10.35–12.92 (11.85 ± 0.68) μm, P/E = 0.98–1.12 (1.04 ± 0.04) and R. mucronata with 12.37–18.49 (15.77 ± 1.58) × 11.35–18.27 (15.69 ± 1.73) μm, P/E = 0.96–1.10 (1.01 ± 0.04) which shows slight differences in size as stated by Mao et al. (2012) in Southern China and Mohd-Arrabe’ & Noraini (2013) in Peninsular Malaysia.

As mentioned by Mao et al. (2012), Mohd-Arrabe’ & Noraini (2013), and Dalimunthe et al. (2023) in South Sumatra coastal area, Indonesia, the pollen shapes of Rhizophora apiculata are prolate, spheroidal or prolate spheroidal. However, from our study, the pollen shapes can be prolate spheroidal or oblate spheroidal.

According to Mao et al. (2012), the pollen grains of Rhizophora apiculata are mostly tricolporate, sometimes tetracolporate; however, in this study, we only found the tricolporate and is consistent with Mohd-Arrabe’ & Noraini (2013) and Dalimunthe et al. (2023).

Conclusions

We have extensively examined specimens with updated morphological descriptions and an identification key for two species of Thai Rhizophora: R. apiculata and R. mucronata. Three names in Rhizophora, are lectotypified: R. apiculata and two associated synonyms of R. mucronata (i.e., R. latifolia and R. macrorrhiza). R. longissima, a synonym of R. mucronata, is considered as a neotype.

Rhizophora is distinguished from other related genera (i.e., Bruguiera, Ceriops, and Kandelia) by its stems with numerous branched stilt roots; narrowly conical, sessile colletters; leaves with numerous, scattered tiny black cork warts on the abaxial surface; mucronate leaf apex; leaf scars with many tiny vascular bundle strands, arranged in a crescent moon; 4-merous flowers; unlobed petals, without appendages; stamens 2–3 times the number of petals; introrse, multi-locellate anthers, dehiscing with an adaxial valve; and half-inferior, 2-locular ovaries, two ovules per locule.

Rhizophora apiculata and R. mucronata are confined to mangrove forests, gregarious on the deep soft mud of estuaries, especially along riverbanks or creeks flooded by high tides, where they are often the dominant species, and are distributed in the south-western, the central, the south-eastern, and the peninsular regions of Thailand.

The thick cuticles, sunken stomata, large hypodermal cells, and cork warts are adaptive anatomical features of leaves in Rhizophora that live in high-salinity habitats, the mangrove environments.

The pollen morphology of Thai Rhizophora species is similar or shows slight differences in size which do not provide good characters for identification within the genus.

In Thailand, Rhizophora apiculata and R. mucronata are often planted for their wood and are used for firewood and for making charcoal.

Findings of this study (taxonomy, leaf and wood anatomy, and palynology) will provide a basis for further development of the other sciences, for conservation and sustainable uses of these mangrove species.

Supplemental Information

Supplemental Information 1 Names of Herbaria.

We would like to thank the curators and staff of the following herbaria BK and BKF for their assistance during visits and allowing access to the herbarium specimens, and those included in the virtual herbaria of AAU, BM, BR, CAL, E, G, K (including K-W), L (including U), P, US, W, and the Wallich Catalogue Online. We are grateful to the plant collectors of these two Rhizophora species and herbaria: Naturalis Biodiversity Center, Leiden, the Netherlands (L); Conservatoire et Jardin botaniques de la Ville de Genève, Genève, Switzerland (G); and National Museum of Natural History (NMNH), Smithsonian Institution, Washington, District of Columbia (D.C.), U.S.A. (US) for the photographs of the lectotypes. We would like to thank Chatree Maknual (Director) and Nawin Phormsin (Forestry Technical Officer, Professional Level) of International Mangrove Botanical Garden Rama IX, Department of Marine and Coastal Resources for their facilitation and kind help in the field work. Special thanks go to Wanwisa Bhuchaisri for the line drawings. We also would like to thank Pimrak Aksornwachpalin, Rattikarn Nuroon, Kittiya Rungruangkasem, and Yarttra Ritthiwan, undergraduate students in the Department of Botany, Faculty of Science, Kasetsart University and Dr Piyangkun Lueangjaroenkit, a lecturer in the Department of Microbiology, Faculty of Science, Kasetsart University for their kind help with field work.

Additional Information and Declarations

Competing Interests

Author Contributions

Data Availability

The authors declare that they have no competing interests.

Chatchai Ngernsaengsaruay conceived and designed the experiments, performed the experiments, analyzed the data, prepared figures and/or tables, authored or reviewed drafts of the article, and approved the final draft.

Pichet Chanton performed the experiments, prepared figures and/or tables, authored or reviewed drafts of the article, and approved the final draft.

Nisa Leksungnoen performed the experiments, authored or reviewed drafts of the article, and approved the final draft.

Suwimon Uthairatsamee performed the experiments, authored or reviewed drafts of the article, and approved the final draft.

Nittaya Mianmit performed the experiments, authored or reviewed drafts of the article, and approved the final draft.

The following information was supplied regarding data availability:

The raw data is available in the tables and the additional specimens examined.

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
