# Peer review of "A taxonomic revision of Rhizophora L. (Rhizophoraceae) in Thailand"

_PeerJ, doi:10.7717/peerj.17460_

## Round 0.1 · original submission · Major Revisions

Dear Dr. Ngernsaengsaruay,

The reviewers checked your manuscript. They detected some weaknesses that must be addressed. In particular, they consider the impact and novelty of the research within the broader context of current research in botany or conservation is not clear. They also find the novelty of this research not clear in the light of the fact that the genus has already been revised for the Flora of Thailand. So, in the current shape, the manuscript looks like a repetition of the previous taxonomic revision. Please therefore clearly identify the knowledge gaps that the research is addressing in the Introduction. Please, respond point-to-point to the comments of reviewers to speed up the process of revision. Once again, thank you for submitting your manuscript to PeerJ and we look forward to receiving your revision.
Sincerely,
Gabriele Casazza

Reviewer 1 ·

Basic reporting

No comment

Experimental design

1. it could further clarify the significance of this revision in the broader context of mangrove conservation and biodiversity research.
2. The ethnobotanical data presented in the research are insightful. Enhancing this section with detailed methodologies for data collection and informant selection criteria would improve its reliability and depth. A concise overview of the informants' expertise or experience with the species would also be valuable.
3. For an even stronger replication framework, the manuscript could benefit from including more explicit details on any statistical analyses used in the morphological differentiation of species

Validity of the findings

1. The manuscript does not explicitly assess the impact and novelty of its findings within the broader context of current research in botany or conservation. While the taxonomic revision of Rhizophora species in Thailand is undoubtedly valuable, the manuscript could be improved by explicitly stating its novel contributions to the field, such as new morphological insights, conservation implications, or the revision's role in facilitating future research on mangrove biodiversity and ecology.
2. It would be beneficial to detail any statistical analyses conducted, particularly in comparing morphological data across species and regions.
3. The conclusions of the manuscript effectively summarize the findings from the taxonomic revision, including the morphological distinctions between Rhizophora species and their distribution within Thailand. The conclusions are directly linked to the research question, focusing on the taxonomic clarification of Rhizophora species. Nonetheless, the manuscript could enhance its conclusions by discussing the implications of these findings for conservation strategies and future research directions, thus directly tying the results back to broader ecological and conservation concerns

Additional comments

More explicitly stating potential future research directions arising from this study's findings, particularly in areas like genetic diversity, ecological function, and conservation management, could provide valuable insights for the scientific community and encourage meaningful replication and extension of the work.

·

Basic reporting

The manuscript is written in mostly correct, professional English. There are however few errors, e.g.

Lines 36-38, abstract: poor English language, the pollen grains are not ‘ … tricolporate apertures’, but either have apertures, or are tricolporate. Likewise e.g. line 912-913, ‘the pollen apertures of Rhizophora apiculata are mostly tricolporate’ - no, the pollen grains are tricolporate, not the apertures. Please verify the terminology.
Line 66: ‘categorized as exclusive species that are’ - somewhat poor English, perhaps better ‘categorized as species that are exclusively limited…’’
Lines 100-101: The sentence is not finished, something is missing, perhaps the word ‘although’ is confusing and should be ‘previously’, then it should be correct.

Literature references are quite sufficient (although the web resource of JSTOR Global Plants (http://plants.jstor.org’ should be considered, which is important for type specimens not included in the cited online herbarium websites). However some references are not cited correctly:
- ‘Office of the Forest Herbarium, Forest and Plant Conservation Research Office, 1129 Department of National Parks, Wildlife and Plant Conservation. 2014‘
should instead be cited as ‚Pooma R, Suddee S [Eds] (2014) Tem Smitinand's Thai Plant Names, revised edition 2014.’ (this was decided by the Flora of Thailand Editorial Board’ and should be followed).
- In several references the citation of publication place and name of publisher is confused, e.g.
Blume (1827, 1847), Leiden: Lugduni Batavorum [Lugduni Batavorum is the Latin name of Leiden];
Chantrapornsyl and others: not ‘Thailand’ but a city, also Gledhill D. 2002, where UK is a country, not a city.
- Hou D. 1970. Rhizophoraceae. In: Smitinand T, Larsen K, eds. Flora of Thailand. Vol. 2(1). Bangkok: Tistr Press, 5-15 - published by the Forest Herbarium, we need publisher not printed.
- Griffith 1836: Is a journal not a book

Article structure and illustrations are excellent, certainly a strength of this paper.

Line 45: Rhizophoracaceae, ending in ‘eae’, are Latin plural, so ‘Rhizophoracaceae are’, not ‘is'. Its a common mistake.

lines 785-786: sentence incomplete.

TAXONOMY:
There are a number of errors in type citations, and for a ‘taxonomic’ revision with new lectotypes included this really should not be.
The cited type of Rhizophora is a lectotype (not mentioned! - see Index Nominum Genericorum website). The weblink of Flora of Pakistan (line 263 and others) should be updated (see TROPICOS).
The lectotype of R. apiculata had already been annotated as such presumably by Ding Hou, although apparently not published. Better mention it.
lines 362 ff.: E. conjugata sensu Arnott (1838) is a nom. illeg., but in the protologue Arnott includes the name and type of C. candelaria DC. It is therefore a nomenclatural, homotypic synonym of R. candelaria, and this important information is missing. The discussion and the additional references in lines 364-369 are superfluous and a bit confusing, the nomenclature is more important and it is a synonym by definition.
lines 497-498: the Lamarck herbarium has the acronym P-LA (see Index Herbariorum). Did the authors contact the curator of Paris? It would be useful to learn if there is a type specimen or if it is absent. The type locality is Mauritius, formerly called Ile-de-France. Why syntype, only one specimen was cited?
lines 504-506, 594 ff.: Blanco’s species was described in 1837, Merrill’s Species Blancoanae collected c. 1915, so the latter was not present in 1837, is therefore not original material (Art. 9.4 ICN) and cannot be lectotype (lectotypes can only be chosen from among the original material). Species Blancoanae are usually considered as neotypes (Art. 9.13), not lectotypes. See https://www.biodiversitylibrary.org/bibliography/2116
lines 511-512: what is ‘2, n.v.’? Hochreutiner types should be in G, where he worked. Perhaps contact the curators of G. You have so many new data on anatomy etc., so why note update the data on the presence of type specimens also?
lines 589-593: The original types of Griffith are in K, and K has a type sheet (see JSTOR Plants). You should better select this, or at least mention that you know that there is a duplicate. ‘nor did he mention the name of the herbarium where the material was housed’ - for this information see TL-2 online.
Line 603: HB = Herbarium Buitenzorg, now Bogoriense, where he was curator, NOT ‚Herb. Lugd. Bat.’ [which refers to L].

Experimental design

Standard herbarium and anatomical methods were applied correctly, and the manuscript contains original primary research.

Validity of the findings

This submission is entitled ‘A taxonomic revision of Rhizophora in Thailand’. The genus has already been revised for the Flora of Thailand before, and it should be mentioned at the beginning of abstract and introduction that this new revision confirms the treatment for the Flora of Thailand, i.e, there are no taxonomic novelties now (no new data), but new details are provided about the known species, nomenclature (i.e. types), distribution and anatomy. The Flora of Thailand should be acknowledged somewhat more more. What is new and valuable in this submission is less the taxonomy itself but additional data not published before.
Except for this, previous findings are included and cited sufficiently, and the conclusions are reliable and well documented.

Additional comments

It is a very useful and nicely illustrated manuscript. Just the typifications and references would need more careful revision, in addition to some minor details.

Reviewer 3 ·

Basic reporting

Basic reporting is fine. No comment.

Experimental design

The study was sufficiently thorough, and the author's keen efforts in this work are apparent. However, the research question was not well defined, relevant or meaningful. The knowledge gap that the research is addressing is not identified. Rhizophora apiculata and R. mucronata are two of the most easily identified mangrove species, thus the necessity of a taxonomic revision was not fully demonstrated in the manuscript. The author should explain why a detailed taxonomic revision of Thai Rhizophora is needed. Are the Rhizophora species in Thailand botanically unique in any way?

Also, how does this relate to Rhizophora species found in neighbouring countries? while comparing the various botanical feature of Thai specimens to those in previous studies, it is important to note where the previous studies were conducted, in order to understand the geographical relevance of these differences.

Validity of the findings

The rationale and benefit to the literature has not been demonstrated clearly.

Minor comment: It would be better to cite Tomlinson 2016 (second edition) instead of Tomlinson 1986 (first edition).

---

## Round 0.2 · Minor Revisions

Dear Dr. Ngernsaengsaruay,
the manuscript was strongly improved according to the reviewer’s suggestions. However, few points still need to be improved and/or clarified. I didn’t find straight responses to some points of reviewers 3. In particular: 1) how does this relate to Rhizophora species found in neighbouring countries? 2) while comparing the various botanical feature of Thai specimens to those in previous studies, it is important to note where the previous studies were conducted, in order to understand the geographical relevance of these differences. Please, respond to these points. Moreover, at the end of the Introduction you said that the manuscript includes distributional maps of species. However, I didn’t find them neither in the figures nor in supplementary files. Please, respond to these points. Once again, thank you for submitting your manuscript to PeerJ and we look forward to receiving your revision.
Sincerely,
Gabriele Casazza

---

## Round 0.3 · accepted · Accept

Dear Dr. Ngernsaengsaruay,

You have satisfied all the reviewer's concerns. So, I am pleased to inform you that your paper " A taxonomic revision of Rhizophora L. (Rhizophoraceae) in Thailand" is accepted for publication in the PeerJ. Congratulations!
Thank you for submitting your work to PeerJ.

Sincerely,
Gabriele Casazza